# Unsupervised Radar Range-Azimuth Super-Resolution via LiDAR Guided Diffusion Prior

## Abstract

The angular resolution of automotive radar is fundamentally limited by the Rayleigh criterion and the number of antennas, resulting in sparse radar point clouds that hinder high-precision perception. Recent approaches address this limitation by training discriminative or conditional generative models on paired radar-LiDAR data, learning an explicit mapping from radar measurements to LiDAR-like outputs. However, these methods are sensitive to calibration errors and fail when LiDAR data is unavailable. In this work, we propose the first *cross-modality, unsupervised* radar point cloud super-resolution method by formulating radar enhancement as a Bayesian inverse problem. We explicitly model the radar sensing process with a differentiable forward operator, defining the measurement likelihood, and combine it with a LiDAR-trained latent diffusion model that captures the distribution of physically plausible point clouds. During inference, we perform posterior sampling that integrates this generative prior with the radar likelihood, yielding reconstructions that are both geometrically rich and strictly measurement-consistent — without requiring paired radar-LiDAR data. Experiments on the RADIal dataset show that our approach matches the performance of supervised mapping-based methods, while generalizing significantly better to unseen datasets such as K-Radar. These results demonstrate the effectiveness of combining physics-based modeling with distributional priors, offering a robust and practical solution for radar perception in real-world deployments. Our code is available at https://anonymous.4open.science/r/RadarINV2025.

## 1 Introduction

Radar is an essential sensing modality for industrial automation (Palffy et al., 2020; Liu et al., 2023) and autonomous driving (Zheng et al., 2023; Huang et al., 2023) due to its robustness under adverse weather (Yang et al., 2024) and low-light conditions. However, its angular resolution is fundamentally limited by the Rayleigh criterion, which is determined by the operating wavelength and the number of antenna elements. As a result, only reliable peak-retention methods can be applied to extract valid points, which inherently leads to sparse radar point clouds

Traditional approaches improve angular resolution through array signal processing techniques such as MUSIC (Schmidt, 1986) and ESPRIT (Roy & Kailath, 1989), which exploit signal correlations but remain constrained by hardware aperture size and noise sensitivity. To overcome these limits, recent works leverage cross-modal supervision by aligning radar with high-resolution LiDAR, using either discriminative neural networks (Jin et al., 2023; Kim et al., 2024) or conditional generative models (Wu et al., 2024; Zhang et al., 2024; Luan et al., 2024) to map radar data to LiDAR-like outputs. While effective, these mapping-based methods depend on paired radar–LiDAR data, which is expensive to collect, sensitive to calibration errors, and unavailable in many deployment scenarios where only radar is present.

In this paper, we address this limitation by removing the need for paired radar–LiDAR data and instead formulating radar enhancement as a Bayesian inverse problem. Our approach is motivated by the observation that a generative prior trained on LiDAR captures rich geometric statistics of real-world scenes, providing a powerful representation of plausible point cloud distributions. By formulating radar super-resolution under a Bayesian posterior sampling paradigm, we elegantly decouple radar enhancement from direct LiDAR supervision, injecting LiDAR knowledge into the

enhancement process as a distributional constraint rather than an explicit mapping. This yields enhanced radar point clouds that exhibit the fine-grained spatial structure characteristic of LiDAR data while remaining strictly consistent with the observed radar measurements. In doing so, our method avoids the need for paired and calibrated radar–LiDAR data, offering a practical and robust solution for real-world deployment.

Our contributions are threefold:

- We introduce the first unsupervised radar point cloud enhancement method that leverages a LiDAR-trained diffusion prior without requiring paired data.

- We formulate radar enhancement as a Bayesian posterior sampling problem, combining the radar forward model as likelihood with the generative diffusion prior.

- We demonstrate that our approach achieves comparable quality to supervised methods on RADIal and exhibits superior generalization on the unseen K-Radar dataset.

## 2 RELATED WORKS

**Traditional Radar Super-Resolution:** Classical methods such as MUSIC (Schmidt, 1986) and ESPRIT (Roy & Kailath, 1989) improve angular resolution by exploiting the covariance structure of received signals. While effective in ideal conditions, they remain limited by the Rayleigh criterion and degrade under low SNR or with closely spaced targets.

**Mapping-based Radar Enhancement:** To surpass these physical limits, researchers have trained neural networks to map radar data to LiDAR-like dense outputs. Some works use 2D or 3D radar point clouds as inputs to CNN or UNet architectures (Jin et al., 2023; Kim et al., 2024), while others exploit richer radar signal representations such as range-Doppler or range-azimuth maps (Cheng et al., 2021; Prabhakara et al., 2023). Although these discriminative methods increase point cloud density, they require large amounts of paired data and generalize poorly to new domains.

Generative approaches, including conditional GANs and diffusion models, learn a mapping from radar to LiDAR distribution, producing higher-quality and more diverse outputs (Ho et al., 2020; Rombach et al., 2022; Wu et al., 2024). Some works introduce additional consistency constraints (Zhang et al., 2024) or biased domain mappings (Luan et al., 2024) to improve fidelity. However, these approaches still rely on radar-LiDAR pairs for training, making them unsuitable for unpaired or LiDAR-absent scenarios.

**Mapping-free Radar Enhancement:** An alternative is to treat radar enhancement as an inverse problem and use generative models as priors rather than direct mappers. This paradigm has been successfully applied in various fields like imaging (Chung et al., 2023), medical reconstruction (Luo et al., 2020), and signal restoration (Lemercier et al., 2024). In these applications, the objective is to find a high-fidelity output that not only satisfies the physical measurements but also conforms to a pre-defined prior distribution. Early approaches relied on explicit regularization priors, such as $L_1/L_2$ (Shkvarko et al., 2016) regularization for sparsity and smoothness, or Total Variation (TV) (Rudin et al., 1992) for preserving sharp edges. More recently, powerful generative models like Diffusion Models have been used to solve these problems through posterior sampling (Chung et al., 2023; Song et al., 2024; Rout et al., 2024). However, these works typically operate within a single modality, where both the prior and the measurement lie in the same domain.

In contrast, we introduce a cross-modality formulation: our prior is learned entirely from LiDAR data, yet we apply it to reconstruct radar point clouds by combining it with the radar forward model in a Bayesian posterior sampling framework. This enables radar super-resolution without paired training data and represents, to our knowledge, one of the first demonstrations of diffusion-guided inverse problem solving across sensing modalities. Crucially, by bridging physics-based radar modeling with LiDAR domain priors, our approach achieves high-resolution, geometrically consistent radar outputs while maintaining strong generalization to unseen radar datasets.

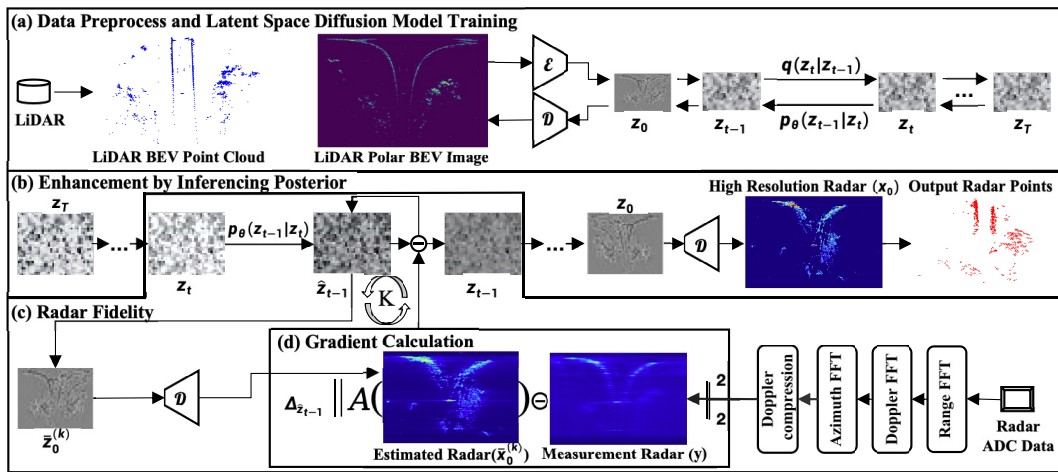

Figure 1: **Diffusion-prior-based cross-modality radar enhancement.** Let $\mathbf{z}_t$ denote the latent at diffusion step $t$, with $\mathcal{E}$ and $\mathcal{D}$ as the encoder and decoder of the VQ-VAE. The forward and reverse diffusion processes are modeled by $q(\mathbf{z}_t \mid \mathbf{z}_{t-1})$ and $p_\theta(\mathbf{z}_{t-1} \mid \mathbf{z}_t)$, respectively. $\mathcal{A}(\cdot)$ denotes the radar forward operator, and $\ominus$ indicates element-wise subtraction. **(a)** A latent diffusion model is trained on LiDAR latents to learn an unconditional prior $p(\mathbf{z}_0)$. **(b)** At inference, we draw samples from the Bayesian posterior $p(\mathbf{z}_0 \mid \mathbf{y})$ by combining the learned prior with the radar measurement likelihood. **(c)** Each reverse diffusion step alternates between prior-guided denoising and an $L_2$ measurement-consistency update. **(d)** The measurement gradient is computed between $\mathcal{A}(\bar{\mathbf{x}}_0^{(k)})$ and the observed radar measurement $\mathbf{y}$, and backpropagated to update $\widehat{\mathbf{z}}_{t-1}$. Steps (b)–(d) constitute the inference process, which requires no further training beyond the prior learned in step (a).

## 3 PROPOSED METHOD

### 3.1 METHOD OVERVIEW

We formulate radar point cloud enhancement as a Bayesian inverse problem that integrates the physics of radar sensing with a learned LiDAR prior. Let $\mathbf{y}$ denote the observed radar measurement (e.g., range–azimuth heatmap), and let $\mathbf{x}$ represent the unknown high-resolution radar point cloud we wish to recover. The radar sensing process is modeled by a forward operator $\mathcal{A}(\cdot)$, yielding

$$\mathbf{y} = \mathcal{A}(\mathbf{x}) + \mathbf{n}, \tag{1}$$

where $\mathbf{n} \sim \mathcal{N}(0, \sigma^2 \mathbf{I})$ is measurement noise. Recovering $\mathbf{x}$ from $\mathbf{y}$ constitutes an ill-posed inverse problem due to the underdetermined nature of radar sensing (few antennas vs. many potential reflectors).

Rather than learning a direct mapping from radar to LiDAR, we seek the *posterior distribution*

$$p(\mathbf{x} \mid \mathbf{y}) \; \propto \; p(\mathbf{y} \mid \mathbf{x}) \, p(\mathbf{x}), \tag{2}$$

where $p(\mathbf{y} \mid \mathbf{x})$ is the likelihood induced by the radar measurement model and $p(\mathbf{x})$ is a prior capturing the distribution of dense, geometrically plausible point clouds. We instantiate $p(\mathbf{x})$ using a *latent diffusion model* trained solely on LiDAR data. During inference, we draw samples from the posterior equation 2 by iteratively combining the diffusion prior gradient (denoising step) with the measurement-consistency gradient (data-fidelity step), producing a radar enhancement that is both physically consistent and LiDAR-like in structure. Figure 1 summarizes this process.

### 3.2 RADAR FORWARD MODEL

We model the radar angle measurement following a narrowband plane-wave assumption. Consider a uniform linear array with $N$ receive antennas spaced at distance $d$. For a reflector at azimuth angle $\theta$, the phase delay between adjacent antennas is given by

$$\Delta\phi = \frac{2\pi d}{\lambda} \cos\theta, \tag{3}$$

where $\lambda$ is the radar wavelength. The complex baseband signal received by the $k$-th antenna can be written as

$$s_k(t) = s_0(t)e^{-jk\Delta\phi}. \tag{4}$$

Stacking responses from $M$ reflectors yields the array observation model

$$\mathbf{y} = \mathbf{A}(\boldsymbol{\theta}) + \mathbf{n}, \tag{5}$$

where $\boldsymbol{\theta} = [\theta_1, \ldots, \theta_M]^\top$ are unknown angles of arrival, and $\mathbf{A}(\boldsymbol{\theta}) = [\mathbf{a}(\theta_1), \ldots, \mathbf{a}(\theta_M)]$ is the steering matrix with $\mathbf{a}(\theta)$ denoting the array steering vector, $\mathbf{n}$ is noise. Extending this derivation over range bins yields the range–azimuth measurement function $\mathcal{A}(\mathbf{x})$ used in equation 1. Because $M \gg N$, equation 5 is underdetermined, motivating a regularized or Bayesian reconstruction approach.

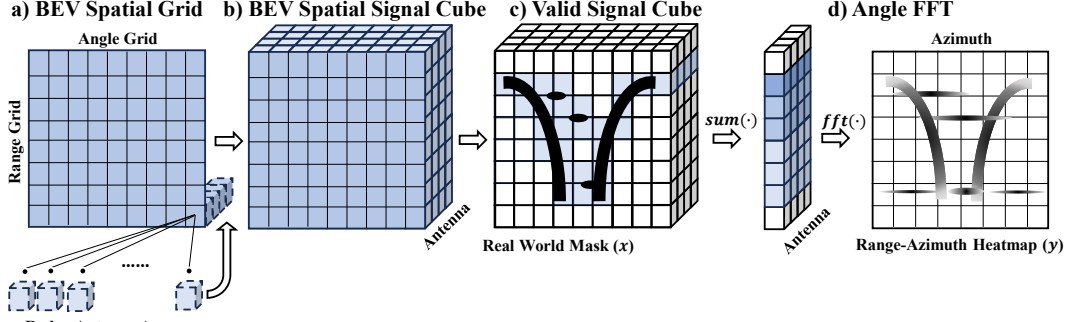

Figure 2: Parallel radar forward model.

However, a direct implementation of this process is typically discrete, posing a challenge for gradient-based optimization in frameworks like PyTorch. To enable end-to-end training and ensure our method is computationally feasible, we developed a parallel radar forward model. The key idea is to treat the the unknown range-azimuth real-word geometry $\mathbf{x}$ as a mask. Then use this mask to filter the spatial signal cube and acquire range-azimuth heatmap sensing results of radar. This process is described in Figure 2.

At the step a), a BEV spatial grid is initialized as the target sensing space for radar sensor. The space is under polar coordinate. Suppose there is an object in each spatial grid that can reflect microwave electromagnetic wave to the radar sensor, the signal on each radar antenna is computable theoretically. Then in b), for each spatial grid, a BEV spatial signal cube that consisted of range, angle and antenna is formed. However, the signal matrix represents full space signal, which is impossible because the target space could not be full filled with objects. Therefore, the real world mask $\mathbf{x}$ is used to filter out valid signal grid in c). Then in step d), after summation of the complex signal from different valid grids on the same antenna, the range-antenna signal matrix is constructed. Finally, after angle FFT, the range-azimuth heatmap $\mathbf{y}$ result of radar imaging process is given. It is worthy to mention that, because the $\mathbf{x}$ is an unknown which needed to be solved, it is randomly initialized and then updated with the solving process.

### 3.3 BAYESIAN FORMULATION

To put the forward model in a Bayesian framework, we assume a Gaussian noise model for the radar observations following classical statistical processes theory. Assuming i.i.d. Gaussian noise, the likelihood function derived from equation 5 is

$$p(\mathbf{y} \mid \mathbf{x}) = \mathcal{N}(\mathbf{y}; \mathcal{A}(\mathbf{x}), \sigma^2\mathbf{I}). \tag{6}$$

Maximum-likelihood estimation would minimize $\|\mathbf{y} - \mathcal{A}(\mathbf{x})\|_2^2$ but is prone to noise amplification and overfitting. Classical $L_1/L_2$ regularization imposes sparsity or smoothness on $\mathbf{x}$ but does not capture the complex spatial statistics of real-world scenes. We therefore introduce a **learned prior** $p(\mathbf{x})$ based on a latent diffusion model, which encourages $\mathbf{x}$ to follow the LiDAR point cloud distribution and thus remain dense and geometrically coherent.

## 3.4 Diffusion Prior and Posterior Sampling

We adopt a two-stage latent diffusion prior: (i) a VQ-VAE encoder–decoder $(E, D)$ compresses LiDAR BEV images into latent variables $\mathbf{z}_0 = E(\mathbf{x}_0)$, and (ii) a denoising diffusion model learns the score function of the latent distribution. The forward diffusion process adds Gaussian noise:

$$\mathbf{z}_t = \sqrt{\bar{\alpha}_t}\,\mathbf{z}_0 + \sqrt{1 - \bar{\alpha}_t}\,\boldsymbol{\varepsilon}, \quad \boldsymbol{\varepsilon} \sim \mathcal{N}(0, \mathbf{I}), \tag{7}$$

and a neural network $\boldsymbol{\epsilon}_\delta(\mathbf{z}_t, t)$ is trained to predict $\boldsymbol{\varepsilon}$ by minimizing

$$\mathcal{L}_\delta = \left\| \boldsymbol{\varepsilon} - \boldsymbol{\epsilon}_\delta(\mathbf{z}_t, t) \right\|_2^2. \tag{8}$$

During inference, we perform reverse diffusion steps starting from $\mathbf{z}_T \sim \mathcal{N}(0, \mathbf{I})$. At each step, the denoised latent is updated by combining: (a) the diffusion prior gradient (denoising step) and (b) the measurement-consistency gradient enforcing $\mathcal{A}(D(\mathbf{z}_0)) \approx \mathbf{y}$. This yields a MAP-consistent estimate of $\mathbf{z}_0$, which is then decoded as $\hat{\mathbf{x}} = D(\mathbf{z}_0)$ to obtain the enhanced radar point cloud.

---

**Algorithm 1** Posterior Sampling with Diffusion Prior

---

**Require:** Radar measurement $\mathbf{y}$, forward model $\mathcal{A}$, trained diffusion prior $\boldsymbol{\epsilon}_\delta$, step size $\zeta$, measurement update steps $K$, measurement scale $\gamma$
**Ensure:** Enhanced radar point cloud $\hat{\mathbf{x}}$
1:  Draw $\mathbf{z}_T \sim \mathcal{N}(\mathbf{0}, \mathbf{I})$
2:  **for** $t = T, T - 1, \ldots, 1$ **do**
3:      $\boldsymbol{\mu}_t \leftarrow \boldsymbol{\epsilon}_\delta(\mathbf{z}_t, t)$
4:      $\hat{\mathbf{z}}_{t-1} \leftarrow \mathbf{z}_t + \lambda_{\text{diff}} \boldsymbol{\Sigma}_t^{-1}(\boldsymbol{\mu}_t - \mathbf{z}_t)$
5:      **for** $k = 1$ to $K$ **do**
6:          $\mathbf{x}_0^{(k)} \leftarrow D(\hat{\mathbf{z}}_{t-1})$
7:          $\mathcal{L}_{\text{meas}} \leftarrow \|\gamma \mathbf{y} - \mathcal{A}(\mathbf{x}_0^{(k)})\|_2^2$
8:          $\hat{\mathbf{z}}_{t-1} \leftarrow \hat{\mathbf{z}}_{t-1} - \zeta \nabla_{\hat{\mathbf{z}}_{t-1}} \mathcal{L}_{\text{meas}}$
9:      **end for**
10:     $\mathbf{z}_{t-1} \leftarrow \hat{\mathbf{z}}_{t-1}$
11: **end for**
12: $\hat{\mathbf{x}} \leftarrow D(\mathbf{z}_0)$
13: **Return:** $\hat{\mathbf{x}}$

---

Algorithm 1 summarizes the inference process for radar enhancement. Starting from a Gaussian noise latent $\mathbf{z}_T$, we perform reverse diffusion over $T$ steps. At each step $t$, the diffusion prior $\boldsymbol{\epsilon}_\delta(\mathbf{z}_t, t)$ produces a denoised estimate with step size $\lambda_{\text{diff}}$, yielding an intermediate latent $\hat{\mathbf{z}}_{t-1}$. To ensure consistency with the observed radar measurement $\mathbf{y}$, we iteratively refine $\hat{\mathbf{z}}_{t-1}$ for $K$ inner-loop updates by backpropagating the measurement residual $\mathcal{L}_{\text{meas}} = \|\gamma \mathbf{y} - \mathcal{A}(D(\hat{\mathbf{z}}_{t-1}))\|_2^2$ with respect to the latent variable. This gradient step shifts the latent toward solutions whose decoded point cloud better matches the radar observation under the forward model $\mathcal{A}(\cdot)$. After $T$ steps, the final latent $\mathbf{z}_0$ is decoded into the enhanced radar point cloud $\hat{\mathbf{x}} = D(\mathbf{z}_0)$, which is both geometrically plausible (due to the LiDAR-trained prior) and measurement-consistent.

## 4 Experiments and Results

### 4.1 Dataset and Evaluation Metrics

**Dataset**: To evaluate the effectiveness of our proposed method, two autonomous driving sensing datasets, RADIal dataset (Rebut et al., 2022) and K-Radar dataset (Paek et al., 2022), are selected. The RADIal dataset is a collection of 2 hours of raw data from synchronized automotive-grade sensors (camera, laser, High Definition radar) in various environments (city 30.2%, countryside 50.0%, highway 18.2%). The K-Radar dataset is a large-scale 3D perception benchmark for autonomous driving, featuring 35,000 frames of 4D Radar tensor (4DRT) data with power measurements across Doppler, range, azimuth, and elevation dimensions. In our experiments, the K-Radar dataset is only used for cross dataset validation.

**Evaluation Metrics**: Based on previous research (Zhang et al., 2024; Wu et al., 2024; Luan et al., 2024), Chamfer Distance (CD) is primarily used in our experiments to evaluate the mutual minimum distance between generated 3D points and ground truth 3D points. This metric assesses the similarity of two point sets, with smaller CD values indicating higher similarity. In addition to Chamfer Distance, we also consider other metrics such as Unidirectional Chamfer Distance (UCD), Modified Hausdorff Distance (MHD), Unidirectional Modified Hausdorff Distance (UMHD), and Fréchet Inception Distance (FID).

Table 1: **Parameters of model in different datasets**

| | | | RADIal | K-Radar |
|---|---|---|---|---|
| Training | First Stage (VQ-VAE) | Original LiDAR Size | $512 \times 768 \times 3$ | - |
| | | Encoder/Decoder Layers | 5 | - |
| | | Latent Embedded Dimension | 4 | - |
| | | Latent Embedded Channels | 2048 | - |
| | | Training Batch Size | 2 | - |
| | Second Stage (Latent Diffusion) | Latent LiDAR Size | $32 \times 48 \times 4$ | - |
| | | Encoder/Decoder Layers | 3 | - |
| | | Training Batch Size | 16 | - |
| Inference | | Input Radar Size | $512 \times 768 \times 3$ | $512 \times 768 \times 3$ |
| | | DDPM Steps | 1000 | 500 |
| | | Measurement Scale ($\gamma$) | 1.0 | 1.0 |
| | | Measurement Step Size ($\zeta$) | 0.001 | 0.0005 |
| | | Measurement Step Number ($K$) | 20 | 5 |

**Implementation Details**: Our method trains a diffusion model exclusively on LiDAR data only (e.g., LiDAR point cloud in RADIal dataset), employs the trained model to reconstruct LiDAR style output as prior information during inference procedure, to generate the posterior high resolution radar point cloud by applying the prior on top of likelihood generated by radar heatmap input (e.g., radar heatmap in RADIal, K-Radar or any other dataset). Once trained, the diffusion model which generates the prior can be applied with any other radar data input for inference using our proposed approach.

a) **Pre-processing**: RADIal dataset LiDAR points are first projected into a polar coordinate BEV image with a resolution of $512 \times 768$, covering an angular range of $[-90°, 90°]$ (512 divisions) and a radial range of $[0, 103]$ m (768 divisions) as the same field of view (FoV) and resolution as radar range-azimuth heatmap. The created LiDAR BEV image is a binary mask, with occupied grids set to 1 and empty grids to 0. For radar data, the RADIal dataset API decodes the range-azimuth heatmap from the raw signal matrix into a $512 \times 768$ image, with values normalized to $[0, 1]$ via linear scaling. Similarly, the K-Radar dataset's range-azimuth signals are used, with a Field-of-View from $-53°$ to $53°$ in azimuth and a detection range of $[0, 118]$ m. The K-Radar data is interpolated from its original $256 \times 128$ size to match the $512 \times 768$ resolution of the RADIal dataset. No additional training is required for K-Radar dataset.

b) **Model Training**: The training process begins with a diffusion model trained exclusively on LiDAR data of RADIal dataset, serving as the prior. Two stages are included. The first stage is training a VQ-VAE on an Nvidia L40 GPU with 48 GB of CUDA memory. The training parameters are detailed in Table 1. The LiDAR BEV images are duplicated into 3 channels to give full play to the feature capture capability of VAE. Once the VQ-VAE converges after 62 epochs, the second stage—training the latent diffusion model—commences. This model reaches convergence after 130 epochs.

c) **Model Inference**: For RADIal dataset, radar data is used as input measurement $\boldsymbol{Y}$, refining the $32 \times 48 \times 4$ latent variable $\boldsymbol{z}$ together with diffusion prior gradient. The final output is obtained after 1000 iterative steps and decoding $\boldsymbol{z}$. Table 1 specifies the parameter values for the RADIal dataset: $\zeta = 1.0$, $\gamma = 0.001$, and $K = 20$. For the K-Radar dataset, $\zeta = 1.0$, $\gamma = 0.0005$, and $K = 5$. The previous diffusion prior is used repetitively.

d) **Post-processing**: Once the output has converged, an average is computed across the three channels, which suppresses $512 \times 768 \times 3$ to $512 \times 768$. A threshold of 0.01 is then applied to identify valid BEV grid points for generating the final radar points. After extracting the grid, the output is converted from polar to Cartesian coordinates.

## 4.2 RESULTS

### 4.2.1 RADAR FORWARD MODEL SIMULATION

To illustrate the effectiveness of the parallel radar forward model, the LiDAR data is used as the real work mask **x**. The results are showed in Figure 3. Calibration information of RADIal dataset

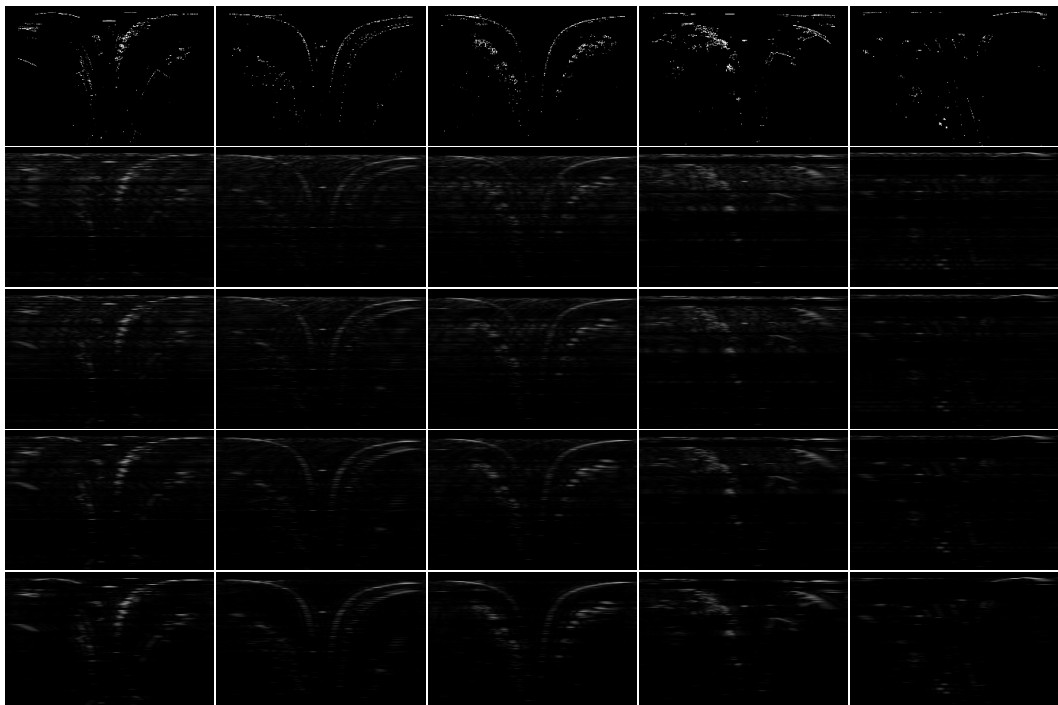

Figure 3: Visualization Results of Calibrated Forward Model. We use LiDAR as input x of forward model then obtain the degraded results y. Each column corresponds to a different frame. The first row shows the input LiDAR point cloud, while the subsequent rows present the outputs of the forward model under different antenna configurations: 8, 32, 64 and 192 antennas. The results reveals that the calibrated forward model produces noticeably sidelobes and interference compared to the ideal theoretical model. This behavior more closely reflects the characteristics of real radar sensing systems.

is considered in forward model to correct the inaccuracy of theoretical signal matrix. Different antennas number setting corresponding to different radar hardware setting. From the visualization results in Figure 3, the fact is known that the parallel radar imaging likelihood function is able to degrade real world x. Less number of antennas means more degradation. Therefore, when solving the radar point cloud enhancement inverse problem based on this likelihood function, one possible solution as x that hidden in this function is available.

### 4.2.2 PERFORMANCE EVALUATION OF RADIAL DATASET

Statistical performance analysis on the RADIal dataset can demonstrate the effectiveness of the proposed method. Table 2 presents the performance metrics of point cloud enhancement results for different methods. Unsupervised methods include CFAR, $L_1$ Reg and $L_2$ Reg. CFAR is a traditional peak extraction method. The other two are methods that originally solved radar angle estimation as an inverse problem. RadarHD and Diffradar are two typical supervised approaches that have recently been proposed. The first is a discriminative model using UNet (Ronneberger et al., 2015), while the second is a generative model which is based on conditional-DDPM (Ho et al., 2020). Notably, there are many works in each category, but only RadarHD is open source. We reproduced Diffradar's data.

It can be observed that our method achieves a superior performance than the supervised learning method, RadarHD. Compared to Diffradar, the point clouds generated by our method exhibit good similarity and density. Figure 4 shows the results of various radar super-resolution point cloud generation schemes. Compared to the sparse results of CFAR, other schemes provide denser point clouds. However, the $L_1/L_2$ regularization schemes introduce a significant number of noise points. For the RadarHD, the main issue is the complete loss of generated details. For Diffradar, while the model indeed has high quality point cloud generation capabilities, it suffers from deficiencies in

Table 2: Statistical analysis of point enhancement on RADIal datasets.

| Methods | Supervised Training | $FID_{BEV}$ ↓ | CD ↓ | UCD ↓ | MHD ↓ | UMHD ↓ |
|---|---|---|---|---|---|---|
| CFAR | ✗ | 242.17 | 6.38 | 3.11 | 2.06 | 1.94 |
| L1 Reg (Shkvarko et al., 2016) | ✗ | 208.17 | 5.69 | 2.39 | 3.36 | 3.12 |
| L2 Reg (Shkvarko et al., 2016) | ✗ | 204.18 | 5.63 | 2.55 | 3.60 | 3.32 |
| RadarHD (Prabhakara et al., 2023) | ✓ | 174.90 | 6.50 | 1.87 | 3.05 | 1.47 |
| Diffradar (Wu et al., 2024) | ✓ | 138.07 | 5.64 | 2.25 | **1.60** | **0.97** |
| Ours | ✗ | **122.35** | **4.08** | **1.47** | 1.88 | 1.14 |

density. In contrast, our method generates point clouds with higher density than the CFAR method, lower noise levels than the $L_1/L_2$ regularization, more detailed point clouds than RadarHD, and higher fidelity than Diffradar.

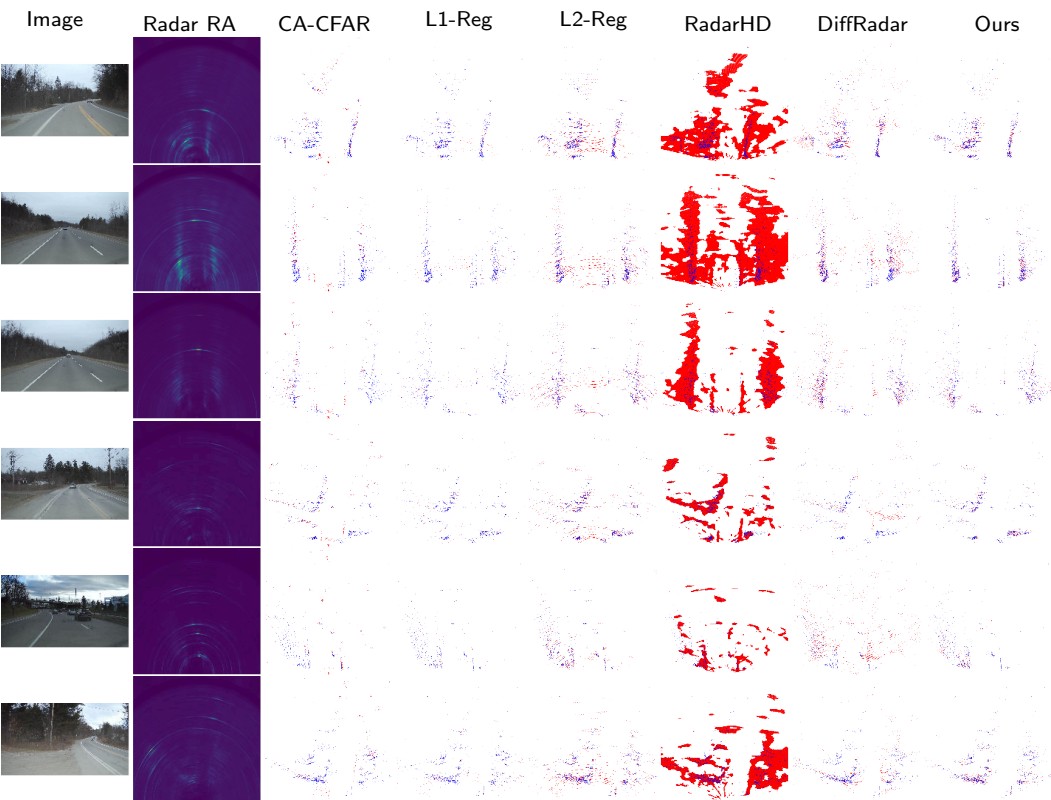

Figure 4: **Qualitative comparison of the RADIal dataset across different methods**. Different randomly selected frames from three scenarios are displayed in rows, while point enhancement results from different methods are shown in columns. In each sub-figure, blue points represent the LiDAR ground truth, while red points indicate the enhanced point cloud outputs.

### 4.2.3 CROSS-DATASET VALIDATION ON K-RADAR DATASET

To illustrate the generalization ability of our method, we directly use radar data from K-Radar dataset as input for each method; no more LiDAR data from K-Radar dataset is included. The inference parameters are listed in Table 1. The statistical and visualization results are depicted in Table 3 and Figure 5 respectively.

It is evident that supervised methods like RadarHD and DiffRadar produce overly dense outputs that saturate the detection space, highlighting their limitations in handling cross-dataset generalization. Similarly, $L_1/L_2$ regression-based methods yield noisy results, while traditional CFAR approaches produce sparse outputs with side-lobe noise. In contrast, our method more accurately recovers the basic structure of the environment, despite missing some regions detected by LiDAR. For instance, as shown in the first row of the figure, points along the left roadside edge and at farther distances

Table 3: **Cross dataset validation on K-Radar dataset**. Our method demonstrates strong generalizability compared to other approaches.

| Method | Supervised Training | $FID_{BEV}\downarrow$ | CD↓ | UCD↓ | MHD↓ | UMHD↓ |
|---|---|---|---|---|---|---|
| CFAR | ✗ | 244.55 | 26.09 | 13.27 | 22.82 | **135.53** |
| $L_1$ Reg (Shkvarko et al., 2016) | ✗ | 263.17 | 49.84 | 26.99 | 46.99 | 386.02 |
| $L_2$ Reg (Shkvarko et al., 2016) | ✗ | 258.69 | 33.34 | 26.94 | 42.46 | 386.60 |
| RadarHD (Prabhakara et al., 2023) | ✓ | - | - | - | - | - |
| Diffradar (Wu et al., 2024) | ✓ | - | - | - | - | - |
| Ours | ✗ | **230.67** | **23.42** | **7.38** | **16.94** | 361.91 |

are absent due to weak signals in the radar input and occlusion caused by the radar's low mounting position on the K-Radar vehicle. Additionally, recovery is constrained by our use of only 16-layer LiDAR as a prior, while K-Radar employs a higher-resolution 64-layer LiDAR, limiting the achievable output fidelity.

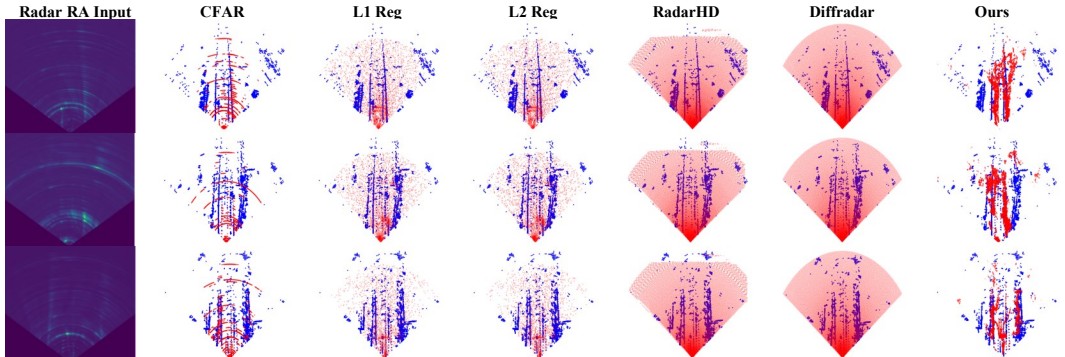

Figure 5: **Cross-dataset enhancement on K-Radar**. Rows show radar inputs; columns show outputs from different methods. Blue and red points denote LiDAR (GT) and generated points respectively. Other methods struggle with noise and domain shifts, while our approach yields cleaner, robust results without requiring paired training data. More results please refer to Figure 10 in Appendix I.

### 4.2.4 CROSS SCENARIO VALIDATION ON RADIAL DATASET

The generalization capability of our method is validated through cross-scenario testing on the RADIal dataset, which includes three scenarios: City (30.2%), Countryside (50%), and Highway (18.8%). Table 4 presents the cross-scenario performance using CD metrics. Training the diffusion prior with all RADIal LiDAR data yields better overall performance across scenarios (CD: 6.31). However, when training solely on Countryside LiDAR data, the model achieves superior performance in the Countryside scenario (CD: 5.85) but underperforms in City and Highway scenarios, leading to a higher overall CD (6.59). The reason is that the diffusion prior is specifically tailored to the Countryside LiDAR data distribution, capturing its unique spatial patterns more effectively. Despite this, the model's

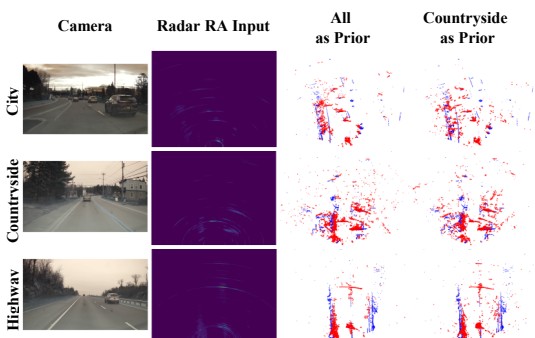

Figure 6: **Visualization results of cross-scenario validation on RADIal dataset**. Three rows demonstrate three scenarios provided by RADIal dataset. In each scenario, two frames are selected. Results of all as prior and countryside as prior are provided. More results please refer to Figure 11 in Appendix I.

Table 4: **Cross scenario validation on RADIal dataset using CD metrics**. The diffusion prior is trained under LiDAR data from Countryside and All of RADIal dataset. Then radar data from different scenarios (City, Countryside and Highway) is used as our model's input.

| Prior | Test Scenario (CD) | | | |
|---|---|---|---|---|
| | City (30.2%) | Countryside (50.0%) | Highway (18.8%) | All |
| Countryside | 7.46 | **5.85** | 7.48 | 6.59 |
| All | **6.77** | 6.26 | **6.01** | **6.31** |

generalization ensures effective point cloud enhancement even without prior data from City and Highway scenarios.

Figure 6 provides a visual comparison of output results across various data subsets. When relying solely on countryside LiDAR data as a prior, the enhanced radar point clouds show increased noise in City and Highway settings, but superior performance in the Countryside itself. Among the different scenarios, the highway environment delivers the strongest performance owing to its simpler layout, whereas the city scenario yields somewhat less precise results due to its intricate urban landscape and varied object distributions, which significantly affect radar input signals.

## 5 CONCLUSION AND FUTURE WORK

This paper proposes a radar point cloud enhancement algorithm using a diffusion model as a prior. Bayesian inference for the radar enhancement is formulated, and the samples are drawn from the posterior distribution using the diffusion model as a prior and the radar imaging equation as a constraint. Through parameter analysis and comparative experiments, the effectiveness and high performance of our method are demonstrated. However, challenges such as the high cost of inference time and the alignment of multi-modal features persist, and will be examined and addressed in future work.

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

## A   PRELIMINARIES: RADAR SENSING AND NOTATION

This section provides a concise overview of FMCW radar processing for a broad audience and introduces the notation used throughout our derivations.

**Range FFT.** Each chirp produces a fast-time signal whose beat frequency encodes target distance. A 1D FFT along this dimension yields the *range spectrum* with range bins indexed by $r$.

**Doppler FFT.** Across $K$ chirps, the slow-time phase progression reflects relative radial velocity. A 1D FFT along the chirp dimension gives the *Doppler spectrum*, forming range–Doppler cells $(r, d)$.

**Angle FFT.** A uniform linear receiver array exhibits angle-dependent phase differences. A 1D FFT across the antenna dimension produces the *angle spectrum*, yielding range–azimuth cells $(r, \alpha)$, which form the range–azimuth heatmap used as the radar measurement.

**Radar data cube.** Stacking the three FFT-domain representations yields

$$\text{RadarCube} \in \mathbb{C}^{R \times D \times N},$$

where $R$ is the number of range bins, $D$ the Doppler bins, and $N$ the number of (virtual) antennas.

**CFAR peak extraction.** CFAR estimates a local noise floor and retains only statistically significant peaks. Although widely used, CFAR removes most non-peak cells, making radar point clouds sparse. Crucially, CFAR and peak-picking are *non-differentiable* and therefore cannot be embedded inside our forward operator.

**Notation.** We denote by $x \in \mathbb{R}^{H \times W}$ the unknown range–azimuth occupancy mask (our reconstruction variable), by $y$ the observed range–azimuth heatmap, and by $A(\cdot)$ the differentiable radar forward model such that

$$y = A(x) + n,$$

with $n$ denoting measurement noise.

## B   MASKING STRATEGY AND COMPLEXITY

To enable end-to-end optimization, the forward model must be differentiable and computationally tractable. We achieve this by expressing the unknown scene $x$ as a range–azimuth occupancy mask applied to a pre-computed spatial response tensor.

**Masked forward operator.** Let

$$S \in \mathbb{C}^{N \times H \times W}$$

denote the spatial response tensor, where $S_{n,h,w}$ is the complex return at antenna $n$ from a unit reflector located at grid cell $(h, w)$. The masked forward model is:

$$V_{n,h,w} = S_{n,h,w}\, x_{h,w}, \qquad u_{h,w} = \sum_{n=1}^{N} V_{n,h,w}, \qquad y_{h,w} = \text{FFT}_{\text{angle}}(u_{h,\cdot})\,[w].$$

All operations (elementwise multiply, antenna summation, FFT) are differentiable, enabling gradient backpropagation through $A(\cdot)$ during posterior sampling.

**Complexity.** The masked operator requires

$$\mathcal{O}(NHW + HW \log W)$$

per forward pass, with efficient fused GPU kernels.

**Alternative schemes.** A dense simulation that individually accumulates contributions from all spatial cells to all azimuth bins scales as

$$\mathcal{O}(NH^2W^2),$$

which is impractical for iterative sampling. CFAR-based pipelines involve thresholding and peak-picking, which are non-differentiable and therefore unusable inside $A(\cdot)$.

**Conclusion.** The masking strategy is required to (i) keep the forward model differentiable and (ii) reduce computational cost to a tractable regime for posterior sampling.

## C  PROOF OF ANGLE ESTIMATION USING A DIFFUSION MODEL PRIOR

To solve the problem depicted in equation 2, with a diffusion model, the posterior distribution is expressed as:

$$p(\boldsymbol{x}|\boldsymbol{y}) \propto p(\boldsymbol{y}|\boldsymbol{x})p_\delta(\boldsymbol{x}). \tag{9}$$

To reduce GPU memory usage, a latent diffusion model is applied, reformulating the problem in the latent space:

$$p(\boldsymbol{z}|\boldsymbol{y}) \propto p(\boldsymbol{y}|\mathcal{D}(\boldsymbol{z}))p_\delta(\boldsymbol{z}), \tag{10}$$

where $\boldsymbol{z}$ is the latent representation of $\boldsymbol{x}$, and $\mathcal{D}(\cdot)$ is the decoder. During each iteration, the update follows:

$$p(\boldsymbol{z}_{t-1}|\boldsymbol{z}_t, \boldsymbol{y}) \propto p(\boldsymbol{y}|\mathcal{D}(\hat{\boldsymbol{z}}_0))p_\delta(\boldsymbol{z}_{t-1}|\boldsymbol{z}_t), \tag{11}$$

where $\hat{\boldsymbol{z}}_0$ is an estimate of $\boldsymbol{z}_0$, since $\boldsymbol{x}_0 = \mathcal{D}(\boldsymbol{z}_0)$, but $\boldsymbol{x}_t \neq \mathcal{D}(\boldsymbol{z}_t)$. Thus, the measurement model gradient must be computed using $\hat{\boldsymbol{z}}_0$.

To maximize equation 11), we apply Maximum A Posteriori (MAP) estimation. The logarithmic posterior is:

$$\log p(\boldsymbol{z}_{t-1}|\boldsymbol{z}_t, \boldsymbol{y}) \propto \log p(\boldsymbol{y}|\mathcal{D}(\hat{\boldsymbol{z}}_0)) + \log p_\delta(\boldsymbol{z}_{t-1}|\boldsymbol{z}_t). \tag{12}$$

Differentiating and setting to zero gives:

$$\nabla_{\boldsymbol{z}_{t-1}} \log p(\boldsymbol{z}_{t-1}|\boldsymbol{z}_t, \boldsymbol{y}) = 0, \tag{13}$$

yielding:

$$\nabla_{\boldsymbol{z}_{t-1}} \log p(\boldsymbol{Y}|\mathcal{D}(\hat{\boldsymbol{z}}_0)) + \nabla_{\boldsymbol{z}_{t-1}} \log p_\delta(\boldsymbol{z}_{t-1}|\boldsymbol{z}_t) = 0. \tag{14}$$

The diffusion model gradient is:

$$\nabla_{\boldsymbol{z}_{t-1}} \log p_\delta(\boldsymbol{z}_{t-1}|\boldsymbol{z}_t) = \nabla_{\boldsymbol{z}_{t-1}} \left( -\frac{1}{2}(\boldsymbol{z}_{t-1} - \boldsymbol{\mu}_\delta(\boldsymbol{z}_t, t))^\top \boldsymbol{\Sigma}_\delta(\boldsymbol{z}_t, t)^{-1}(\boldsymbol{z}_{t-1} - \boldsymbol{\mu}_\delta(\boldsymbol{z}_t, t)) \right)$$
$$= -\boldsymbol{\Sigma}_\delta(\boldsymbol{z}_t, t)^{-1}(\boldsymbol{z}_{t-1} - \boldsymbol{\mu}_\delta(\boldsymbol{z}_t, t)), \tag{15}$$

where $\boldsymbol{\mu}_\delta(\boldsymbol{z}_t, t)$ is predicted by the neural network $\epsilon_\delta(\cdot)$. The measurement model gradient is:

$$\nabla_{\boldsymbol{z}_{t-1}} \log p(\boldsymbol{y}|\mathcal{D}(\hat{\boldsymbol{z}}_0)) = \nabla_{\boldsymbol{z}_{t-1}} \|\boldsymbol{y} - A(\mathcal{D}(\hat{\boldsymbol{z}}_0))\|_2^2. \tag{16}$$

Since only $\boldsymbol{z}_t$ is available at step $t$, Tweedie's formula (Song et al., 2024) estimates $\hat{\boldsymbol{z}}_0$:

$$\hat{\boldsymbol{z}}_0 = \frac{1}{\sqrt{\alpha_t}} \left( \boldsymbol{z}_t - \sqrt{1 - \alpha_t} S_\delta(\boldsymbol{z}_t, t) \right). \tag{17}$$

The diffusion prior gradient ensures the output aligns with the LiDAR data domain, while the measurement gradient regulates fidelity to the radar measurement $\boldsymbol{y}$. These gradients are iteratively applied to an initial noise input, with step and weight parameters optimized for performance, as detailed in Algorithm 1.

## D  DERIVATION DETAILS.

We provide the derivations requested by the reviewer, including assumptions and approximations.

### D.1  ARRAY OBSERVATION MODEL

From Eq. (4) to Eq. (5): Array Observation Model Under the narrowband plane-wave assumption, the signal at the $k$-th antenna is:

$$s_k(t) = s_0(t)\, e^{-jk\Delta\phi},$$

where $\Delta\phi = \frac{2\pi d}{\lambda} \cos\theta$ is the inter-antenna phase shift.

Assumptions:

- Superposition of multiple reflectors.
- Far-field approximation (common phase progression).
- Narrowband model (constant amplitude across antennas).

$$\mathbf{y} = \sum_{m=1}^{M} \mathbf{a}(\theta_m) + \mathbf{n} = \mathbf{A}(\boldsymbol{\theta}) + \mathbf{n},$$

which is Eq. (5).

### D.2 RANGE–AZIMUTH FORWARD OPERATOR

To extend the array model from Eq. (5) to range–azimuth sensing as in Eq. (1), we incorporate:

- range discretization via the range FFT,
- azimuth mapping via angle FFT,
- spatial masking via the unknown occupancy $x$.

Let $\mathbf{s}(p)$ denote the simulated antenna response from spatial cell $p = (h, w)$. Summing contributions at each range bin $r$ and applying the angle FFT yields:

$$y(r, \alpha) = \mathrm{FFT}_{\mathrm{angle}} \left( \sum_{p \in \mathcal{S}(r)} x(p)\, \mathbf{s}(p) \right) + n,$$

which forms the differentiable range–azimuth forward operator:

$$y = A(x) + n,$$

corresponding to Eq. (1).

## E    MODEL EFFICIENCY ANALYSIS AND ACCELERATION

Since the project remains in the prototyping phase and has not yet progressed to deployment, raw inference cost appears high. However, extensive research exists on accelerated sampling strategies that can dramatically reduce computation, such as fewer noise steps, higher-order solvers, or alternative diffusion formulations (DDIM, consistency models, etc.).

To illustrate feasible speed-ups with minimal performance degradation, we explore two straightforward techniques: (a) reducing input resolution and (b) early stopping during sampling.

Figure 7 quantifies the performance–latency trade-off, while Figure 8 shows convergence behavior at different numbers of inference steps.

**Effect of Input Resolution.** Figure 7 shows the Chamfer Distance (CD) versus inference latency for three input sizes (512×768, 256×384, and 128×192). Reducing the input size substantially accelerates inference: the 128×192 resolution achieves a **7× speed-up** over the full resolution. The performance degradation is modest. The CD only increases by roughly **1×** when down-sampling from 512×768 to 128×192. This indicates that the model maintains reasonable accuracy under aggressive down-sampling, making low-resolution inference suitable for latency-critical applications.

**Effect of Early Stopping.** To study the convergence of the iterative denoising process, Figure 8 visualizes intermediate predictions at inference steps $\{0, 200, 400, 600, 800, 1000\}$ for the three input resolutions. Three consistent observations emerge: (i) higher resolutions converge more slowly and continue to refine the output beyond step 600; (ii) lower resolutions converge earlier, with only marginal changes after step 600; and (iii) across all resolutions, the results at step 600 already resemble the final output at step 1000. This suggests that the model can be **effectively early-stopped**, reducing inference latency by **40–60%** with minimal visual degradation.

**Summary.** Combining both acceleration strategies reduces the inference latency from **36 minutes to 2 minutes** (approximately **5.5%** of the original cost), while preserving reconstruction fidelity close to the full-resolution, full-iteration baseline. These results demonstrate that the proposed framework can be efficiently deployed under limited computational budgets without significantly compromising quality.

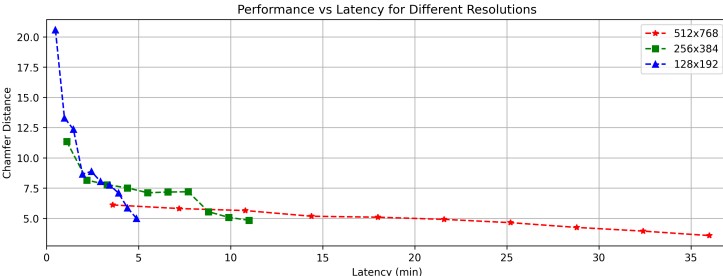

Figure 7: Model efficiency analysis. The x-axis denotes inference time, and the y-axis denotes Chamfer Distance (CD) between generated points and ground-truth LiDAR points. Three input resolutions (512×768, 256×384, 128×192) are evaluated.

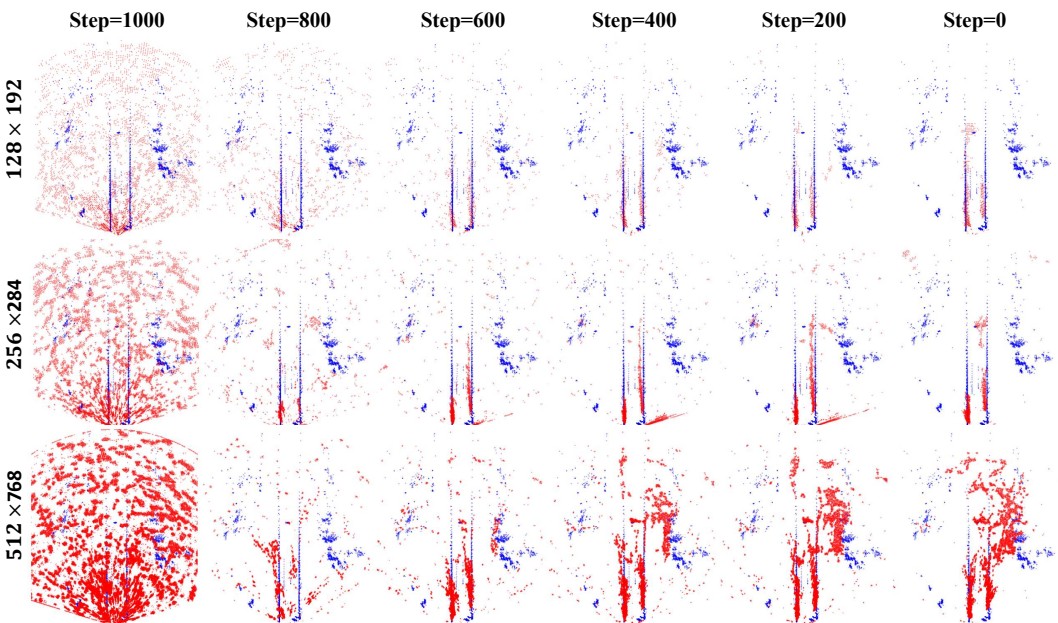

Figure 8: Model convergence visualization at different inference steps (0–1000) under different input resolutions. Higher resolutions converge more slowly, while lower resolutions reach stable predictions earlier, enabling effective early stopping.

## F    EVALUATION OF OBJECT DETECTION.

We evaluate radar enhancement using downstream PointPillars [3] detection, keeping detector training/testing identical across methods. Only the enhanced point clouds differ.

Key configuration of the PointPillars detector is as follows:

- Backbone: MeanVFE + VoxelResBackBone8x
- BEV encoder: HeightCompression (256 ch.)
- Head: CenterHead (Car), stride 8

- Training: 32 batch/GPU, 30 epochs, Adam+OneCycle

The following experiment evaluates how different radar enhancement methods affect downstream 3D object detection performance. We use a standard PointPillars detector trained on each enhanced data. The results, reported as AP at IoU thresholds 0.3/0.5/0.7, are summarized in the table 5.

Table 5: 3D Object Detection Performance (AP) on RADIal Dataset.

| IoU | LiDAR | CFAR | L1-Reg | L2-Reg | RadarHD | Diffradar | Ours |
|-----|-------|------|--------|--------|---------|-----------|------|
| 0.3 | **0.894** | 0.798 | 0.148 | 0.176 | 0.521 | 0.343 | **0.882** |
| 0.5 | **0.876** | 0.601 | 0.110 | 0.106 | 0.314 | 0.240 | **0.874** |
| 0.7 | **0.809** | 0.295 | 0.047 | 0.049 | 0.142 | 0.153 | **0.688** |

Our method transfers geometric fidelity to downstream detection significantly better than other methods and approaches the LiDAR upper bound. Visualization is provided in Figure 9.

## G  ADDITIONAL EXPERIMENT: LAPLACIAN NOISE

We additionally evaluated a Laplacian noise model:

$$p(y \mid x) \propto \exp\left(-\frac{\|Ax - y\|_1}{b}\right), \tag{18}$$

which replaces the L2 consistency with an L1 penalty. Laplacian noise is often used for outlier robustness. The results are summarized below:

Table 6: Comparison of Gaussian vs. Laplacian noise models on RADIal dataset.

| Methods | $\text{FID}_{\text{BEV}} \downarrow$ | CD $\downarrow$ | UCD $\downarrow$ | MHD $\downarrow$ | UMHD $\downarrow$ |
|---------|------|------|------|------|------|
| Ours w/ Gaussian Noise | **122.35** | **4.08** | **1.47** | **1.88** | **1.14** |
| Ours w/ Laplacian Noise | 123.64 | 4.21 | 1.88 | 2.03 | 1.78 |

As shown in Table 6, the performance of the Laplacian noise model is very similar to the Gaussian model across all metrics. While an L1-based consistency term theoretically encourages robustness to sparse outliers, automotive FMCW radar noise is predominantly dense Gaussian thermal and ADC noise, which explains why L2 remains slightly better and more stable. The small numerical differences further indicate that the choice between Gaussian and Laplacian noise does not materially affect our method.

## H  SENSITIVITY TO MEASUREMENT-UPDATE STEPS AND SAMPLING STEPS

We conducted an ablation study to evaluate the sensitivity of sampling quality with respect to the number of measurement update steps (K) and diffusion sampling steps (T). The Chamfer Distance (CD) results are summarized in Table 7. We have two key observations:

Table 7: Sensitivity analysis of measurement-update steps (K) and sampling steps (T) on RADIal dataset using Chamfer Distance (CD).

| T \ K | 20 | 40 | 60 | 80 | 100 | 200 |
|-------|------|------|------|------|------|------|
| T = 50 | 3.05 | 2.48 | 3.15 | 3.21 | 3.44 | 3.23 |
| T = 100 | 3.43 | 3.57 | 3.34 | 3.80 | 3.30 | 3.99 |
| T = 200 | 3.71 | 3.27 | 3.95 | 3.51 | 3.65 | 3.44 |
| T = 500 | 4.90 | 3.34 | 3.47 | 3.77 | 4.46 | 3.42 |
| T = 1000 | 3.80 | 3.89 | 3.93 | 3.76 | 3.64 | 4.11 |

**1. Overall sensitivity is low across a wide range of T and K**

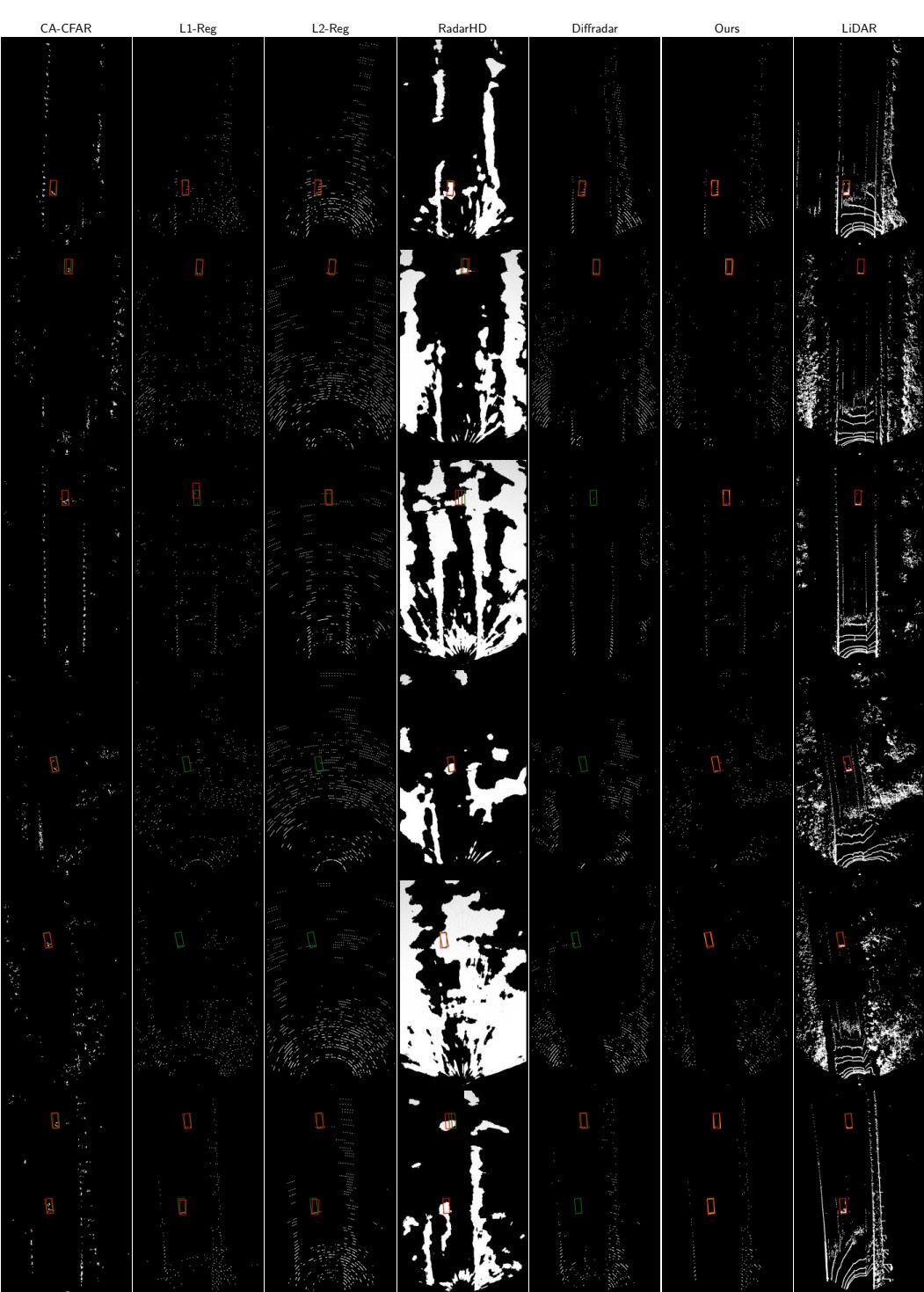

Figure 9: Visualization Results Object Detection on RADIal dataset with Different Radar Enhancement Methods.

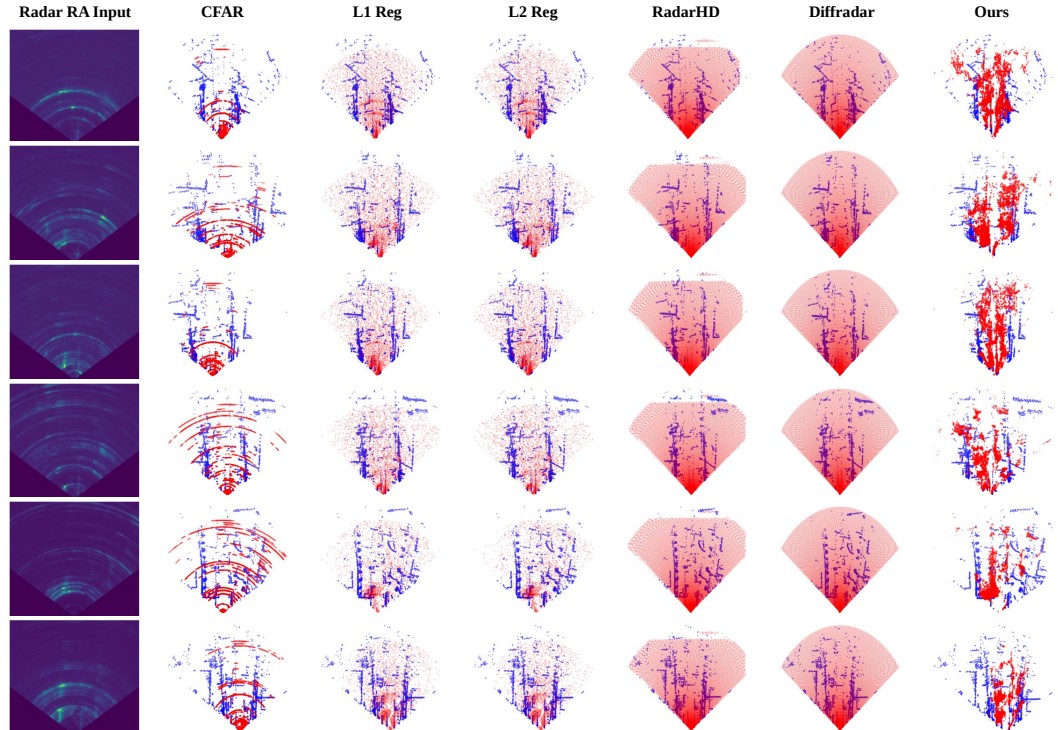

| Radar RA Input | CFAR | L1 Reg | L2 Reg | RadarHD | Diffradar | Ours |
|---|---|---|---|---|---|---|

Figure 10: Cross-dataset comparison of radar point cloud enhancement of different methods on K-Radar dataset.

We evaluated diffusion steps **T** from 50 to 1000 and measurement update steps **K** from 20 to 200. In such test, the original DDIM sampler is used to replace the original 1000 steps DDPM-Solver for providing adjustability of **T**. The results indicate that our method is not highly sensitive to these hyperparameters. Across a very broad sweep ($5\times$ variation in **T** and $10\times$ variation in **K**), the results remain within a relatively narrow performance range (CD $\approx$ 2.4–4.0). Increasing **T** or **K** does not consistently improve reconstruction quality; in many cases, excessively large values even degrade performance. This indicates that the sampling process is not highly sensitive to these hyperparameters, and large values provide limited benefit.

**2. Best trade-off: small T, moderate K**

The best performance is achieved at $T = 50$, $K = 40$ (CD = 2.48). Overall, the region $T = 50 - 100$, $K = 40 - 60$ consistently provides strong reconstruction quality while keeping computation efficient.

Thus, our method remains practical and robust, and a small number of sampling steps with a moderate number of measurement updates offers the best balance between performance and efficiency.

# I   MORE VISUALIZATION RESULTS

We depict more visualization results in this section including point cloud enhancement visualization results of cross-dataset visualization results of K-Radar dataset in Figure 10, and cross-scenario visualization results of RADIal dataset in Figure 11.

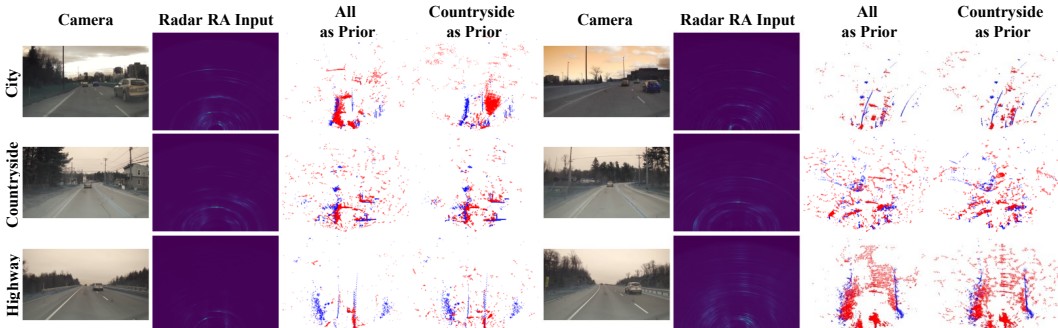

Figure 11: Cross-scenario comparison of radar point cloud enhancement of different methods on RADIal dataset.

## J  LLM USAGE STATEMENT

We used a large language models only to refine grammar and improve the clarity of language in this manuscript. No part of the research ideation, experiment design, or analysis was performed by an LLM.

