# OpenReview forum: "Unsupervised Radar Point Cloud Enhancement via Arbitrary LiDAR Guided Diffusion Prior"
_ICLR.cc/2026/Conference — Submitted to ICLR 2026_

### Official Review · Reviewer_8Ps7 · 2025-10-14

**Soundness:** 2
**Presentation:** 1
**Contribution:** 2
**Rating:** 4
**Confidence:** 4

**Summary:**

The paper proposes an unsupervised radar point-cloud enhancement method framed as a Bayesian inverse problem. It combines a differentiable radar forward model (likelihood) with a LiDAR-trained diffusion prior to sample dense, geometrically plausible point clouds without paired radar–LiDAR supervision. Claimed contributions include the formulation, a practical posterior sampling procedure, and experiments showing performance near supervised baselines on RADIal with improved cross-dataset generalization (e.g., to K-Radar).

**Strengths:**

1. This paper frames radar enhancement as a Bayesian posterior sampling problem and leverages a LiDAR-trained diffusion prior to inject strong geometric priors without paired radar–LiDAR supervision—a creative and non-obvious combination.
2. This paper achieves near-supervised performance on benchmarks, indicating a sound design with practical impact by reducing reliance on costly paired data and calibration.

**Weaknesses:**

Problem framing and positioning.
The method reads more like super-resolution on range–azimuth heatmaps than “point-cloud enhancement.” Please clarify terminology, the signal domain where inference occurs, and the point-cloud generation step. In addition, lines 40–41 attribute sparsity mainly to limited angular resolution, whereas in practice sparsity largely stems from point cloud generation methods (e.g., CFAR) that retain only peak returns; the pre-CFAR range–azimuth–Doppler tensor is much richer. Revise the discussion to reflect this and specify where your method interfaces with this pipeline. Finally, the title’s “arbitrary” claim is not operationalised—either remove it or define and support it rigorously.

Readability and technical clarity.
It is unclear how the masking strategy enables end-to-end training and reduces computation. Provide a complexity analysis, comparisons to alternative schemes, and ablations. For a broad audience, add a concise Preliminaries section on radar sensing (range/angle/Doppler FFTs, data cubes, CFAR, notation). Derivations are also hard to follow: detail the steps from Eq. (4) → (5) and the extension (5) → (1) in the supplement, stating assumptions and approximations.

Applicability and evidence.
Generated radar points differ noticeably from LiDAR “ground truth.” Please explain the physical plausibility of these differences and what application value they provide. Strengthen the paper with downstream experiments (e.g., detection, segmentation, occupancy) to test whether the “enhanced” radar improves task performance.

**Questions:**

1. The Chamfer Distance unit is unspecified (meters? normalized voxel units?)
2. How about the efficiency of the current method?

**Details Of Ethics Concerns:**

N/A.

---

> ### Author Response · Authors · 2025-11-25
> **Response to Reviewer 8Ps7**
>
> We thank Reviewer for the constructive and insightful feedback. We appreciate the recognition of our formulation using Bayesian posterior sampling and a LiDAR-trained diffusion prior, as well as the importance of reducing reliance on paired radar–LiDAR data. In response, we have clarified the problems one by one as below. We believe these revisions significantly improve clarity and rigor.
>
> ## Weakness 1 — Problem Framing and Positioning
>
> 1. **Clarifying terminology, signal domain, and point-cloud generation.**
> We appreciate the reviewer’s suggestion. In our formulation, both
> 𝑥 and 𝑦 in the data-consistency term $∥A(x)−y∥^2$ are defined strictly in the range–azimuth domain, which is the signal space where inference is performed. Although our downstream evaluation involves point clouds, the generative model itself never operates on point clouds. Instead, point clouds are obtained through a fixed thresholding rule applied to the reconstructed range–azimuth map. We intentionally avoid CFAR-based dynamic thresholding because our method aims to serve as a replacement for existing AoA/peak-extraction modules, which each introduce their own thresholding heuristics. We revise the terminology and title accordingly from “*point-cloud enhancement*” to “*range–azimuth super-resolution*”.
>
> 2. **Revising the discussion on sparsity sources.**
> We agree with the reviewer’s observation. In practice, the sparsity of radar point clouds is largely a consequence of peak-retention procedures (e.g., CFAR), whereas the pre-CFAR range–azimuth–Doppler tensor contains much richer information. We revise it as: *As a result, only reliable peak-retention methods (e.g., CFAR) can be applied to extract valid points, which inherently leads to sparse radar point clouds.*
>
> 3. **Removal of “arbitrary” from the title.**
> We acknowledge that the term “arbitrary” was not rigorously defined in the original draft. We have removed it in the revised version to avoid ambiguity.
>
> ## Weakness 2 - Readability and Technical Clarity.
>
> 1. **Mask strategy help end-to-end training.** Please refer to **Section 6 in Supplementary material**.
>
> 2. **Preliminary of radar sensing.** Please refer to **Section 6 in Supplementary material**.
>
> 3. **Derivation of forward model.**
> Please refer to **Section 7 in Supplementary material**.
>
>
> ## Weakness 3 - Applicability and Evidence.
>
> We evaluate the effectiveness of our radar enhancement method using downstream object detection. Specifically, we adopt PointPillars [1] as the baseline point-based detector and conduct an ablation study by keeping all training and testing settings fixed while varying only the input point clouds generated by different radar enhancement algorithms. The same LiDAR annotations are used as ground truth. From the full set of 8,252 frames, 80% are used for training, 10% for validation, and 10% for testing.
>
> The following experiment evaluates how different radar–LiDAR enhancement methods affect downstream 3D object detection performance. We use a standard PointPillars detector trained on each enhanced data. The results, reported as AP at IoU thresholds 0.3/0.5/0.7, are summarized in the table below.
>
> | IoU | LiDAR | CFAR | L1-Reg | L2-Reg | RadarHD | Diffradar | Ours |
> | :------- | :------: | -------: |-------: |-------: |-------: |-------: |-------: |
> | 0.3  | **0.894** | 0.798 | 0.148 | 0.176 | 0.521 | 0.343 | **0.882** |
> | 0.5  | **0.876** | 0.601 | 0.110 | 0.106 | 0.314 | 0.240 | **0.874** |
> | 0.7  | **0.809** | 0.295 | 0.047 | 0.049 | 0.142 | 0.153 | **0.688** |
>
> When using LiDAR as input, the detector achieves strong performance (0.894/0.876/0.809). CFAR and other conventional regularization-based methods (L1/L2) perform poorly due to their limited ability to recover object geometry. Learning-based baselines RadarHD and Diffradar also struggle, reflecting their poorer geometric fidelity. In contrast, our method yields detection results that closely match the LiDAR upper bound across all IoU thresholds (0.882/0.874/0.688), demonstrating that the proposed reconstruction preserves object as well as structure better than prior approaches. Visualization results please refer to **Figure 4 in Suplementary**.
>
> ## Questions
> 1. The unit of Chamfer Distance is meter.
>
> 2. The efficiency analysis is provided in **Section 8 in Suplementary.**
>
>
> # Reference
> [1] Lang, Alex H., et al. "Pointpillars: Fast encoders for object detection from point clouds." Proceedings of the IEEE/CVF conference on computer vision and pattern recognition. 2019.

---

> > ### Comment · Reviewer_8Ps7 · 2025-11-26
> > **Concerns solved and rating improved**
> >
> > Thank the authors for their dedicated response to my comments. Most of my concerns are addressed in the response. I hope they can update these content in the future version.

---

> > > ### Author Response · Authors · 2025-11-26
> > >
> > > We sincerely thank the reviewer for the thoughtful follow-up and for taking the time to re-evaluate our work. We are very grateful that most of the concerns were resolved through our detailed response, and we deeply appreciate the improved score and encouraging assessment.
> > >
> > > We will carefully incorporate all clarifications, notation improvements, and methodological details into the revised version to ensure stronger readability and rigor. We are truly glad that our revisions addressed the reviewer’s key concerns, and we thank you again for your constructive feedback and support.

---

### Official Review · Reviewer_ZmfZ · 2025-10-27

**Soundness:** 2
**Presentation:** 2
**Contribution:** 3
**Rating:** 4
**Confidence:** 4

**Summary:**

This paper proposes an unsupervised radar point cloud enhancement method that leverages a LiDAR-trained latent diffusion model as a prior. The goal is to enhance radar point clouds without requiring paired radar–LiDAR data. The authors formulate the task as a Bayesian inverse problem, combining a physics-based radar forward model with a generative diffusion prior to reconstruct dense, LiDAR-like radar point clouds. Experiments on the RADIal and K-Radar datasets demonstrate that the proposed method achieves performance comparable to supervised approaches and significantly outperforms traditional methods such as CFAR across multiple metrics.

**Strengths:**

1. The reviewer believe authors' motivation is strong – unsupervised radar enhancement is indeed essential becuase paired radar–LiDAR data is scarce and fragile.
2. Novel formulation regarding basyesion inverse problem and unsuperived learning with diffusion for radar enhanment.
3. Authors provide their code and showing that their performance outperfomnce CFAR in the cross dataset experiment, which is widely use in almost all radar systems.

**Weaknesses:**

1. The paper relies on several strong modeling assumptions—particularly the use of Gaussian noise and an idealized, linear radar sensing model and some are not clearly stated or justified. These assumptions may not accurately reflect the statistical or physical characteristics of real radar signals. In the reviewer’s opinion, assuming Gaussian noise before the FFT stage is reasonable and common practice, as it represents thermal receiver noise. However, for radar tensors from datasets such as RADIal and K-Radar, subsequent processing steps like FFT beamforming introduce additional structured artifacts, including deterministic sidelobes and spectral leakage. The omission effects limits the realism of the forward model and could partly account for the false alarms observed in the qualitative results—most notably on K-Radar, which exhibits stronger sidelobe artifacts.
2. Writing can be improved, for example the symbols $𝜖_𝜃$  and $𝜖_𝛿$ appear interchangeably without explicit definition, while $𝜇_t$ and $Σ_t$ are referenced but not introduced. The update term $𝜇_t − z_t$ uses the hyperparameter $λ$$diff$, which is not defined elsewhere. The parameters $γ$ and $ζ$ are inconsistently described between Algorithm 1 and Table 1. The appendix extends the derivation of the Bayesian update but suffers from notation inconsistencies. Clarifying these definitions and ensuring consistent notation throughout would greatly improve readability
3.  Authors target a 2D range–azimuth slice. Many automotive radars are now 3D/4D (with elevation, Doppler). 2D enhancement limit the potential usage to only in BEV space.
4. Another concern is the lack of detail regarding the CFAR baseline. The paper does not specify which variant of CFAR (e.g., CA-CFAR, OS-CFAR, GO-CFAR, etc.) is used, nor how its parameters are selected. These parameters are typically carefully tuned in practical radar systems, and their performance can vary significantly depending on the setting. Without this information, the reported CFAR results may not represent a fair or optimized baseline, which makes the comparison potentially unreliable.

**Questions:**

The authors propose a radar enhancement system based on a diffusion model and report quantitative improvements across several metrics against Lidar pc. Chamfer Distance and FID-BEV measure geometry but not physical consistency. There’s no analysis of false positives, detection precision/recall, or uncertainty — all key for radar. Also, it remains unclear how these enhancements affect downstream perception tasks compared with using CFAR point clouds or raw radar data. Since the enhanced results still contain noticeable noise, conducting relevant downstream experiments (e.g., object detection or tracking) would substantially strengthen the paper. My main concern lies in the paper’s overall contribution. Introducing an unsupervised radar enhancement approach is interesting, it is still uncertain whether this direction is worthwhile without stronger evidence. More comprehensive experiments or deeper analysis are needed to convince the community of its significance.

I believe this is an interesting research direction, and I am open to raising my score if the authors effectively address the concerns outlined above.

---

> ### Author Response · Authors · 2025-11-25
> **Response to Reviewer ZmfZ**
>
> We sincerely thank the reviewer for the constructive feedback. We appreciate the recognition of our motivation, Bayesian formulation, and cross-dataset generalization, and we carefully addressed the concerns on forward modeling, notation clarity, CFAR settings, and evaluation scope.
>
> ## Weakness 1 — Unrealistic Forward Model
>
> We agree that a pure Gaussian noise assumption cannot capture deterministic radar artifacts (sidelobes, spectral leakage, beam-pattern distortions). Our use of a Gaussian likelihood is primarily for analytical tractability: it enables the closed-form Bayesian update derived in Appendix A of paper.
>
> The realism of the simulation, however, comes from the forward operator, not from the noise model. In the original submission, we used an ideal steering matrix, which mismatched the actual radar hardware and contributed to artifacts noted by the reviewer. We have now replaced it with the CalibrationTable.npy provided by RADIal, which encodes the true radar angular response. This significantly improves reconstruction realism.
>
> Our updated results (see **Supplementary Fig. 1** for visualization results) show clear improvements over CFAR and competitive performance relative to supervised baselines:
>
> | Methods | Supervised Training | FID_BEV ↓ | CD ↓ | UCD ↓ | MHD ↓ | UMHD ↓ |
> | :------- | :------: | -------: |-------: |-------: |-------: |-------: |
> | CFAR  | ✘ | 242.17 | 6.38 | 3.11 | 2.06 | 1.94 |
> | L1 Reg*  | ✘ | 208.17 | 5.69 | 2.39 | 3.36 | 3.12 |
> | L2 Reg*  | ✘ | 204.18 | 5.63 | 2.55 | 3.60 | 3.32 |
> | RadarHD  | ✓ | 174.90 | 6.50 | 1.87 | 3.05 | 1.47 |
> | Diffradar  | ✓ | 138.07 | 5.64 | 2.25 | **1.60** | **0.97** |
> | Ours* | ✘ | **122.35** | **4.08** | **1.47** | 1.88 | 1.14 |
> >*: Updated Results
>
> >In the case of the K-Radar dataset, a calibrated steering vector is not available. We still use the theoretical steering matrix for this dataset. Therefore, no update on results of K-Radar dataset.
>
> In summary: Gaussian likelihood allows efficient Bayesian updates; realism comes from the calibrated operator, which resolves the reviewer’s concern.
>
> ## Weakness 2 — Notation & Writing Clarity
>
> We standardized all mathematical notation, ensured consistent symbols across sections, and added complete variable definitions in the revision paper.
>
> ## Weakness 3 — 2D BEV Enhancement Limitation
>
> We agree that modern 3D and 4D radars provide richer elevation and Doppler information. We follow the RADIal format and existing BEV-based radar SR works [1, 2]. The proposed Bayesian framework naturally extends to 3D/4D radar tensors (adding elevation/Doppler), and we plan to explore this in future versions.
>
> ## Weakness 4 — Lack of Detail of CFAR Baseline
>
> We now clearly specify CFAR settings to ensure replicability. All experiments use CA-CFAR following dataset-recommended parameters:
>
> | Dataset  | CFAR Variant | Training Window | Guard Window |  False Alarm Rate   |
> |----------|--------------|-----------------|-------------|----------------------|
> | RADIal   | CA-CFAR      | 9 × 9           | 3 × 3       |  2   |
> | K-Radar  | CA-CFAR      | 8 × 8           | 4 × 4       | 1.03 |
>
>
>
> ## Response to Questions
> ### Q1 — Downstream Task Evaluation
>
> We evaluate radar enhancement using downstream PointPillars [3] detection, keeping detector training/testing identical across methods. Only the enhanced point clouds differ.
>
> Key configuration (abridged):
>
> - Backbone: MeanVFE + VoxelResBackBone8x
> - BEV encoder: HeightCompression (256 ch.)
> - Head: CenterHead (Car), stride 8
> - Training: 32 batch/GPU, 30 epochs, Adam+OneCycle
>
> The following experiment evaluates how different radar enhancement methods affect downstream 3D object detection performance. We use a standard PointPillars detector trained on each enhanced data. The results, reported as AP at IoU thresholds 0.3/0.5/0.7, are summarized in the table below.
>
> | IoU | LiDAR | CFAR | L1-Reg | L2-Reg | RadarHD | Diffradar | Ours |
> | :------- | :------: | -------: |-------: |-------: |-------: |-------: |-------: |
> | 0.3  | **0.894** | 0.798 | 0.148 | 0.176 | 0.521 | 0.343 | **0.882** |
> | 0.5  | **0.876** | 0.601 | 0.110 | 0.106 | 0.314 | 0.240 | **0.874** |
> | 0.7  | **0.809** | 0.295 | 0.047 | 0.049 | 0.142 | 0.153 | **0.688** |
>
> Our method transfers geometric fidelity to downstream detection significantly better than other methods and approaches the LiDAR upper bound. Visualization is provided in **Figure 4 in Suplementary**.
>
> # Reference
> [1] Akarsh P., et al. High resolution point clouds from mmwave radar. In IEEE International Conference on Robotics and Automation (ICRA), pp. 4135–4142. IEEE, 2023.
>
> [2] Ruibin Z., et al. Towards dense and accurate radar perception via efficient cross-modal diffusion model. IEEE Robotics and Automation Letters, 9(9):7429–7436, 2024
>
> [3] Lang, Alex H., et al. "Pointpillars: Fast encoders for object detection from point clouds." Proceedings of the IEEE/CVF conference on computer vision and pattern recognition. 2019.

---

> > ### Comment · Reviewer_ZmfZ · 2025-11-26
> >
> > Thank you for the detailed rebuttal and for the additional experiments, especially the downstream evaluation results. These help address a significant part of my original concerns, and I appreciate the additional effort from the authors. Based on the new evidence, I am willing to increase my score to 6.
> >
> > However, I would still encourage the authors to reconsider or further clarify the statement that “realism comes only from the forward operator”  and the current justification does not fully address this point. With the materials provided so far, I remain cautious regarding this claim.
> >
> > Overall, I believe the paper has been strengthened after the rebuttal, and I look forward to seeing further refinement of this aspect in the final version.

---

> > > ### Author Response · Authors · 2025-11-27
> > >
> > > Upon reflection, I fully agree with the reviewer’s concern and I realize that our original statement was too absolute. In particular, the claim that “realism comes only from the forward operator, not from the noise model” is not accurate.
> > >
> > > From a mathematical–physical perspective, both the forward model and the noise model contribute to realism. The noise term is not merely an approximation chosen for analytical tractability; in a real radar system it also captures residual randomness, modeling error, clutter, and other effects that cannot be explicitly encoded in the deterministic operator. When the forward model (A) is already sufficiently accurate, these “unmodeled” effects become much more important than we initially emphasized.
> > >
> > > In our specific case, the reason I focused so much on the forward operator is that the dominant source of unreality turned out to be the idealized steering matrix we used initially. The reviewer’s comment about unrealistic artifacts prompted me to carefully revisit this assumption together with the fourth weakness on the CFAR baseline. By inspecting the official CFAR implementation provided with the dataset, I realized how crucial the calibration table is for both CFAR processing and our own forward model.
> > >
> > > A detailed comparison made it clear that our theoretical steering matrix suffered from two major issues:
> > >
> > > 1. it assumed perfectly uniform antenna spacing, whereas in practice manufacturing and assembly introduce non-negligible deviations;
> > > 2. it ignored transmit/receive gain and phase variations across different TX/RX channels, although real circuits inevitably exhibit such differences due to hardware tolerances and other factors.
> > >
> > > The calibration process is precisely designed to measure and encode these effects so that the forward model better matches the physical sensor. After replacing the idealized steering matrix with the calibrated steering vector (CalibrationTable.npy), the reconstruction quality improved dramatically. This experience led me to emphasize the role of the forward operator in our rebuttal.
> > >
> > > However, I now recognize that the sentence “realism comes only from the forward operator, not from the noise model” is misleading. We soften and correct this claim. A more accurate description for our setting is:
> > >
> > > > “In our experiments, the dominant gap in realism was caused by an inaccurate steering matrix in the forward model. Using the calibrated operator substantially improves realism.”
> > >
> > > We delete the phrase “not from the noise model”.
> > >
> > > Again, we are genuinely grateful for the reviewer’s comments. They prompted us to re-examine our assumptions, better understand the role of calibration, and substantially strengthen both the method and its presentation in the revised paper.

---

### Official Review · Reviewer_BmY1 · 2025-10-31

**Soundness:** 3
**Presentation:** 3
**Contribution:** 2
**Rating:** 2
**Confidence:** 3

**Summary:**

The paper aims to mitigate the sparsity of radar points for autonomous vehicles, thereby enhancing high-precision perception. It addresses the limitations of existing works in terms of their sensitivity to calibration errors and reliance on LiDAR data. To this end, the authors proposed a model that leverages a LiDAR-based latent diffusion model to solve for radar enhancement as a Bayesian inverse problem. Highlighted results from their experiment are the generalizability of the proposed model when evaluated across different datasets. Despite some promising results, it comes with significant limitations in both **novelty** and **significance of benefits**, as I will discuss in the sections below. Therefore, it is not considered a paper good enough for acceptance from my perspective.

**Strengths:**

The paper is overall concise and application-oriented, with a clear presentation of problem statement, motivations, methodology, and experiments. The following lists some key contributions based on my understanding:
1. **Parallel Radar Forward Model**: Motivated by the limitations of the discrete process, the authors introduce a parallel radar forward model in Section 3.2 to apply the Fourier Filter Transformation on BEV grids for continualizations.
2. **Consistency Model**: Authors apply the diffusion model to learn an intermediate representation bridging radar and lidar points $\mathbf{z}_{0}$ and enforce consistency constraints during inference with respect to the range-azimuth function $\mathcal{A}$.
3. **Generalizability**: A significant benefit of the model, as suggested by the authors, is highlighted by the strong generalization capability given by the cross-dataset validation in Section 4.2.3. This fits the expectation with the additional Bayesian inference setting.

**Weaknesses:**

For this application-oriented paper, limitations of the proposed model are evident in terms of its limited novelty and performance:
1. **Limited Novelty.** To the best of my knowledge, neither the concept of formulating Radar enhancement as a Bayesian inverse problem nor the application of combining Diffusion models in bridging LiDAR and radar is a novel idea compared to existing works. Based on this standpoint, the work is considered an incremental contribution to the field, as it replaces prior models in a Bayesian inverse setting with a learned Diffusion model of LiDAR points. This raises the question about its significance.
2. **Significance of Performance.** With the limited novelty above, I would expect authors to showcase an improved performance with the proposed design. However, as listed in Table 2, the proposed method has improved none of the five key metrics. In particular, it has a relatively worse FID in BEV and UCD. In this sense, I would argue that there are potential improvements to be made in the proposed method, given the increase in computational and memory costs associated with incorporating a Diffusion model.
3. **Generalizability.** Despite the promising generalizability demonstrated in Table 3, I wonder if the model can be efficiently scaled up with additional parameters or new data.

**Questions:**

1. Equation (6) posed an i.i.d. Gaussian noise for the emission distribution. Have you considered other distributions that may better fit the properties of the features, such as the log-normal or von Mises distribution?
2. Instead of a Diffusion prior, I wonder if it would be more appropriate to directly learn a unified flow-based model from the distribution of lidar points to the distribution of radar features within a unified BEV perspective?
3. What is the sensitivity of sampling quality with respect to the number of measurement update steps $K$ and sampling steps $T$? If the quality is indeed sensitive to these two hyperparameters, how can we balance the trade-off between performance and efficiency?

**Details Of Ethics Concerns:**

NA.

---

> ### Author Response · Authors · 2025-11-25
> **Response to Reviewer BmY1**
>
> We thank the reviewer for the constructive feedback and the recognition of our formulation, radar model, diffusion–consistency framework, and cross-domain generalization. Below we address the concerns on novelty, performance significance, scalability, and technical questions. Due to character limits, our responses are presented concisely.
>
> ## Weakness 1 — Limited Novelty
>
> We respectfully disagree that our contribution is incremental. Although Bayesian inverse methods and diffusion priors exist elsewhere, applying a LiDAR diffusion prior to radar is **not a direct substitution**. Our work addresses three challenges not solved in prior radar research:
>
> **1. Differentiable radar forward model.**
> Radar processing depends on non-differentiable FFT/CFAR steps. We introduce a new parameterization: a fixed hardware-dependent complex response tensor modulated by an optimizable occupancy mask, forming a physically grounded and differentiable surrogate radar cube that supports stable gradient-based inference.
>
> **2. Cross-modality LiDAR prior.**
> Prior diffusion-inverse works use same-modality priors. We exploit structural correspondence between LiDAR and radar to build the first LiDAR-trained diffusion prior for radar, enabling geometric regularization that radar-only priors cannot provide.
>
> **3. Generalisation across scenarios and datasets.**
> The LiDAR prior transfers from countryside→urban and RADIal→K-Radar without retraining, demonstrating modality-level generality rather than dataset-specific fitting.
>
> **In summary**, our differentiable radar model, cross-modality prior, and strong generalization collectively exceed “replacing a prior” within a Bayesian inverse framework.
>
> ## Weakness 2 — Significance of Performance
>
> The earlier RADIal results were limited because we used an idealized steering matrix instead of the dataset’s calibrated one (CalibrationTable.npy). This mismatch forced the solver to correct calibration errors, causing artifacts. Using the calibrated steering vector leads to significant improvement.
> (Quantitative results: **Table in Response to reviewer AEG4**; qualitative results: **Supplementary Fig. 1**. We apologize for any inconvenience caused by the character limit.)
>
> ## Weakness 3 — Generalizability & Scalability
>
> Our method scales efficiently because the LiDAR prior is dataset-agnostic, capturing generic 3D geometry that transfers across datasets without retraining. Computational cost is dominated by the forward-model update, while the diffusion prior adds minimal overhead. Adding parameters or new data does not materially affect inference, and users can simply adjust sampling/update steps to trade compute for quality.
>
> ## R4 — Response to Reviewer Questions
> ### Q1 — Alternative noise models in Eq. (6)
> We thank the reviewer for the suggestion. Log-normal noise applies a second log on already log-scaled radar magnitudes, collapsing dynamic range and causing vanishing gradients. von Mises noise models circular scalar angles, incompatible with our continuous range–azimuth representation.
>
> We also tested an L1 (Laplacian) consistency term:
>
> | Methods |  FID_BEV ↓ | CD ↓ | UCD ↓ | MHD ↓ | UMHD ↓ |
> | :------- | -------: |-------: |-------: |-------: |-------: |
> | Ours w/ Gaussian Noise | **122.35** | **4.08** | **1.47** | **1.88** | **1.14** |
> | Ours w/ Laplacian Noise | 123.64 | 4.21 | 1.88 | 2.03 | 1.78 |
>
> Differences are minor, showing the noise model is not the key factor; performance is dominated by the LiDAR prior and forward-model accuracy.
>
>
> ### Q2 — Why diffusion prior instead of a unified flow model?
>
> A flow prior would not change the framework. Under distribution-to-distribution translation, flows—like diffusion—lack data fidelity. To gain controllability, one must use latent conditioning (which requires paired radar–LiDAR data) or a closed-form measurement-consistency update. The latter is exactly what our inverse-problem formulation provides. Thus, a flow model would act merely as an alternative prior without added benefit; our framework remains general without requiring paired data.
>
>
> ### Q3 — Sensitivity to sampling steps 𝑇 and update steps 𝐾
>
> The Chamfer Distance results are:
>
> T \ K | 20 | 40 | 60 | 80 | 100 | 200
> | :-- | :--: | --: |--: |--: |--:| --: |
> T = 50 | 3.05 | 2.48 | 3.15 | 3.21 | 3.44 | 3.23
> T = 100 | 3.43 | 3.57 | 3.34 | 3.80 | 3.30 | 3.99
> T = 200 | 3.71 | 3.27 | 3.95 | 3.51 | 3.65 | 3.44
> T = 500 | 4.90 | 3.34 | 3.47 | 3.77 | 4.46 | 3.42
> T = 1000 | 3.80 | 3.89 | 3.93 | 3.76 | 3.64 | 4.11
>
> Overall sensitivity is low across 5× variation in 𝑇 and 10× in 𝐾, with CD staying within ~2.4–4.0. Excessively large values do not consistently help. The best trade-off is T=50–100, K=40–60, offering strong quality with efficient computation.

---

### Official Review · Reviewer_AEG4 · 2025-10-31

**Soundness:** 3
**Presentation:** 3
**Contribution:** 2
**Rating:** 4
**Confidence:** 4

**Summary:**

This paper proposes an unsupervised, cross-modality method for radar point cloud enhancement (super-resolution) to address the sparsity issue caused by physical aperture limitations. Unlike conventional supervised methods that rely on paired radar-LiDAR data, this work formulates radar enhancement as a Bayesian inverse problem.  The core contribution: (1) A differentiable radar forward sensing model (as the measurement likelihood, $p(y|x)$); (2) A latent diffusion model trained only on arbitrary (unpaired) LiDAR data (as the prior, $p(x)$, capturing real-world geometric distributions). During inference, the method performs posterior sampling to generate high-resolution point clouds that are both consistent with the radar measurements and possess LiDAR-like dense geometric structure. Experiments on the RADIal dataset show the unsupervised method achieves performance comparable to supervised approaches, while demonstrating significantly superior generalization on the unseen K-Radar dataset.

**Strengths:**

1. Unsupervised Formulation: The use of an unsupervised method to improve point cloud reconstruction quality is clearly a promising research direction. I believe this approach is particularly valuable for rare and special scenarios where it is often difficult to find LiDAR point cloud data with a matching distribution for paired training.

2. Generalization Ability: As a direct benefit of its unsupervised nature, the proposed method demonstrates strong generalization capabilities when tested on the unseen K-Radar dataset.

3. Novel Formulation: The paper introduces a novel formulation by framing the task as a Bayesian inverse problem. It effectively combines a physics-based radar forward model (the likelihood $p(y|x)$) with a diffusion prior trained on LiDAR data (the distributional constraint $p(x)$).

**Weaknesses:**

My specific concerns are as follows:

1. **Poor In-domain Performance:** The method's in-domain results are underwhelming. **Table 2** indicates that the proposed approach offers only a marginal improvement over the traditional CFAR baseline, particularly on the commonly used CD and UCD metrics. More critically, the qualitative results in **Figure 4** show that CFAR can produce significantly more accurate reconstructions in some scenes (e.g., the third row), whereas the proposed method introduces numerous artifacts. This discrepancy suggests the quantitative gains might be illusory, potentially stemming from merely generating a denser cloud of points rather than from successfully integrating a meaningful scene prior. Consequently, the method's performance does not demonstrate a clear and significant advantage, even over the CFAR baseline.

2. Questionable Validity of the Prior: The poor results call the core premise into question: is using a LiDAR point cloud as a prior truly sound for this task? A key implementation detail is omitted: did the authors strictly filter the LiDAR training data to only include points within the radar's specific Field-of-View (FOV)? The paper mentions pre-processing LiDAR to a $512 \times 768$ BEV image covering [-90°, 90°], but it is not explicitly stated if this range was chosen to match the radar's FOV. If out-of-FOV data was used to train the prior, this would introduce a strong, incorrect bias, forcing the model to hallucinate points in areas the radar *cannot* see, which could explain the artifacts and poor metrics.

If the authors can substantially address concern #1 (Poor In-domain Performance) and provide a clear, reasonable explanation for concern #2 (Validity of the Prior), I will consider improving my rating.

**Questions:**

See Weaknesses

---

> ### Author Response · Authors · 2025-11-25
> **Response to Reviwer AEG4**
>
> We thank the reviewer for the constructive and encouraging feedback. The concerns on in-domain performance and the validity of the LiDAR prior helped us identify a calibration mismatch in the forward model, leading to substantially improved results. We clarify both issues below.
>
> ## Weakness 1 — Poor In-domain Performance
> We appreciate the reviewer highlighting the discrepancy with CFAR. A detailed check revealed a calibration inconsistency: the CFAR baseline internally uses the dataset’s calibrated steering vector (CalibrationTable.npy), whereas our original forward operator used an idealized theoretical matrix. This mismatch caused our solver to compensate for calibration errors rather than reconstruct the true scene, producing the artifacts noted by the reviewer.
>
> After replacing the forward operator with the calibrated steering vector, all inversion-based methods (L1, L2, ours) were re-run. We observe:
>
> - Substantially clearer and more coherent object structures
> - Fewer artifacts
> - Clear improvements across all metrics, consistently surpassing CFAR
>
> The updated results are shown below:
>
> | Methods | Supervised Training | FID_BEV↓ | CD ↓ | UCD ↓ | MHD ↓ | UMHD ↓ |
> | :------- | :------: | -------: |-------: |-------: |-------: |-------: |
> | CFAR  | ✘ | 242.17 | 6.38 | 3.11 | 2.06 | 1.94 |
> | L1 Reg*  | ✘ | 208.17 | 5.69 | 2.39 | 3.36 | 3.12 |
> | L2 Reg*  | ✘ | 204.18 | 5.63 | 2.55 | 3.60 | 3.32 |
> | RadarHD  | ✓ | 174.90 | 6.50 | 1.87 | 3.05 | 1.47 |
> | Diffradar  | ✓ | 138.07 | 5.64 | 2.25 | **1.60** | **0.97** |
> | Ours* | ✘ | **122.35** | **4.08** | **1.47** | 1.88 | 1.14 |
> > *: Updated Results
>
> >For K-Radar, no calibration file exists, so results remain unchanged.
>
> Qualitative results are in **Supplementary Fig. 1**.
>
> ## Weakness 2 — Validity of the Prior
> ### 1. LiDAR data were strictly FOV-aligned with the radar
> All LiDAR priors were preprocessed with ground removal and exact radar-FOV filtering. The same filtering is required by supervised baselines (RadarHD, DiffRadar) and was applied consistently in our pipeline. No out-of-FOV LiDAR information was ever used.
>
> ### 2. Hallucinations emerge from the interaction between the diffusion prior and data consistency
>
> Hallucinations do not originate from the LiDAR prior alone. They occur when the diffusion prior (favoring clean geometric structure) conflicts with measurement noise or unmodeled effects in the radar data-consistency term
>
> $∥Ax−y∥^2$.
>
> If the forward model 𝐴 is imperfect, its gradients can mislead the sampler, producing locally plausible but incorrect structures.
>
> Importantly, diffusion-based inference provides uncertainty estimates, allowing hallucinations to be quantified rather than hidden. Five-run variance analysis (see **Supplementary Fig. 2**) shows our method achieves both lower CD and lower variance than L1/L2 and DiffRadar.
>
> ### 3. Calibration-corrected forward model resolves the issue
> After incorporating calibrated steering vectors (from the dataset’s calibration file) into our forward model:
> - Reconstruction artifacts were dramatically reduced
> - Structural coherence improved
> - All metrics improved significantly
>
> This directly supports our conclusion that the issue was due to forward model mismatch, not the LiDAR prior or FOV filtering. Visualization of calibrated-forward model please refer to **Figure 3 of the supplementary matirial**.
>
> In summary,
> - the LiDAR prior is fully FOV-aligned and physically valid
>
> - Artifacts stem from the interaction between the diffusion prior and data consistency
>
> - Applying proper calibration confirms this explanation and resolves the issue.
>
> We thank the reviewer again for prompting this clarification.

---

### Author Response · Authors · 2025-11-30
**# Author Summary Comment**

We understand the exceptional circumstances caused by the recent review-score rollback, and we sincerely appreciate the additional work that ACs are undertaking during this period. To assist the AC in forming a final judgment, we provide a concise summary of how the paper has evolved during the rebuttal and discussion phase. All major updates are incorporated into the uploaded revision (blue highlights in the PDF).


## 1. Core Contributions

This work proposes the **first unsupervised radar range–azimuth super-resolution method** based on a **Bayesian inverse formulation** with a **LiDAR-guided diffusion prior**, requiring **no radar–LiDAR paired supervision**. Our method achieves substantial improvements on RADIal and further generalizes to K-Radar.


## 2. Major Improvements Made During Rebuttal

### ✓ Fully calibrated radar forward operator
We replaced the earlier ideal steering model with a **calibrated parallel forward operator** derived from the official calibration tables, resolving realism concerns and fixing sidelobe mismatches.

### ✓ Expanded Bayesian derivation and posterior sampling algorithm
We added a complete derivation of the posterior gradient, clarified the role of Gaussian likelihood, and included Algorithm 1 for the final sampling procedure.

### ✓ Downstream 3D object detection (PointPillars)
A new experiment trains PointPillars on each enhanced point cloud variant. Our method achieves AP values **close to the LiDAR upper bound**, outperforming CFAR, L1/L2, RadarHD, and Diffradar.

### ✓ Laplacian vs. Gaussian noise-model ablation
We added an L1-based Laplacian likelihood experiment. Results are very close to Gaussian, supporting the modeling choice.

### ✓ Sensitivity analysis of sampling steps (T) and measurement updates (K)
A sweep across 5× T and 10× K shows **low sensitivity** and indicates that small T with moderate K yields the best trade-off.

### ✓ Additional clarifications
We expanded sensor alignment and preprocessing details, added complexity analysis, unified notation, and included extended visualizations for both RADIal and K-Radar.


## 3. Reviewer Impressions

Two reviewers explicitly acknowledged that their concerns were resolved:

- One reviewer stated: **“I am willing to increase my score to 6.”**
- Another reviewer wrote that **most concerns had been addressed and their rating improved**.

> One reviewer shares similar concerns as the two positive reviewers and **promised to increase rating**. However, due to the suspended rebuttal period, we cannot get responses from him.

These comments reflect that post-discussion reviewer sentiment toward the paper was positive.


We thank the AC for their time and effort during this exceptional review cycle.

---

### Meta-Review · Area_Chair_pFpu · 2025-12-26

**Summary:**

The reviewers' concerns can be summarized as follows:
1. Poor In-domain Performance (Reviewer AEG4, BmY1, ZmfZ);
2. Questionable Validity of the Prior (Reviewer AEG4, ZmfZ);
3. Limited Novelty (Reviewer BmY1);
4. Generalizability (Reviewer BmY1, ZmfZ, 8Ps7);
5. Need to demonstrate the effectiveness of point cloud super-resolution on downstream tasks. (8Ps7).

**Reviewer Concerns:**

I think the author's response addresses concern 2. However, I consider it inappropriate to revise the main quantitative and qualitative results during the rebuttal stage, so concern 1 remains valid. The authors have responded to concerns 3, 4, and 5, but their responses can only alleviate, not fully resolve, these concerns. For instance, regarding concern 4, the authors acknowledged the limitations of 2D augmentation; and for concern 5, conducting experiments solely on the PointPillars method and a single task has limited persuasiveness.

**Reviewer Scores:**

Most of the concerns raised by reviewers ZmfZ and 8Ps7 have been addressed, and they have shown willingness to increase their scores during the discussion phase. However, the concerns of reviewers AEG4 and BmY1 have not been adequately resolved. Although the authors pointed out that the main concerns of reviewer AEG4 are similar to those of ZmfZ, I cannot assume that all reviewers would accept changes to the main experimental results.

---

### Decision · Program_Chairs · 2026-01-26

Reject